# A Two-Player Resource-Sharing Game with Asymmetric Information

**Mevan Wijewardena ***  **and Michael J. Neely**

Department of Electrical and Computer Engineering, University of Southern California,
Los Angeles, CA 90089-2565, USA; mjneely@usc.edu
\* Correspondence: mpathira@usc.edu

**Abstract:** This paper considers a two-player game where each player chooses a resource from a finite collection of options. Each resource brings a random reward. Both players have statistical information regarding the rewards of each resource. Additionally, there exists an information asymmetry where each player has knowledge of the reward realizations of different subsets of the resources. If both players choose the same resource, the reward is divided equally between them, whereas if they choose different resources, each player gains the full reward of the resource. We first implement the iterative best response algorithm to find an $\epsilon$-approximate Nash equilibrium for this game. This method of finding a Nash equilibrium may not be desirable when players do not trust each other and place no assumptions on the incentives of the opponent. To handle this case, we solve the problem of maximizing the worst-case expected utility of the first player. The solution leads to counter-intuitive insights in certain special cases. To solve the general version of the problem, we develop an efficient algorithmic solution that combines online convex optimization and the drift-plus penalty technique.

**Keywords:** resource-sharing games; congestion games; potential games; worst-case utility maximization; drift-plus penalty method



## 1. Introduction

We consider the following game with two players, A and B. There are $n$ resources, each denoted by an integer between 1 and $n$. Each player selects a resource without knowledge about the other player's selection. The state of the game is described by the random vector $W = (W_1, W_2, \ldots, W_n)^\top$, where $W_k$ is the reward random variable of resource $k$. We assume $W_k$ to be independent random variables for each $1 \le k \le n$, taking non-negative real values. If both players choose the same resource $k$, each gets a utility of $W_k/2$. If they choose different resources $k, l$, they receive utilities of $W_k$ and $W_l$, respectively. It is assumed that the mean and the variance of $W_k$ exist and are finite for each $1 \le k \le n$. Both players know the distribution of $W$. Our formulation allows for an information asymmetry between the players. In particular, $\{1, 2, \ldots, n\}$ can be partitioned into four sets $\{\mathcal{A}, \mathcal{B}, \mathcal{C}, \mathcal{AB}\}$ where only player A observes the realizations of $W_k$ for $k \in \mathcal{A}$, only player B observes the realizations of $W_k$ for $k \in \mathcal{B}$, no player observes the realizations of $W_k$ for $k \in \mathcal{C}$, and both players observe the realizations of $W_k$ for $k \in \mathcal{AB}$.

This game can be used to model different real-world scenarios where the agents have asymmetric information regarding the involved information structure. One classic example is the problem of Multiple-Access Control (MAC) in communication systems. Here, communication channels are accessed by multiple users, and the data rate of a channel is shared amongst the users who select it [1]. A channel can be shared using Time Division Multiple Access (TDMA) or Frequency Division Multiple Access (FDMA), where in TDMA, the channel is time-shared among the users [2,3], whereas in FDMA, the channel is frequency-shared among the users [4]. In both cases, the total data rate supported by the channel can be considered the utility of the channel. The problem of information

asymmetry arises since a user might have precise information regarding the total data rate offered by some channels but not others, and the known channels can be different for different users. On the other hand, the users in such a system cannot be trusted since the system may have malicious users (for instance, jammers) who focus on reducing the data rate available to genuine users.

Modified versions of this game apply to problems in economics. For instance, consider a firm that chooses a market to enter from a pool of market options. The chosen market may also be chosen by another firm. The reward of a market is the revenue it brings. Assume a simplified model where there exists a total revenue for each market, and the total revenue is divided equally among the firms entering the market. A reward known to all firms can be considered public information, while a reward known only to one firm is private information of that firm.

The game defined above can be viewed as a stochastic version of the class of games defined in [5], which are resource-sharing games, also known as congestion games. In resource-sharing games, players compete for a finite collection of resources. In a single turn of the game, each player is allowed to select a subset of the collection of resources, where the allowed subsets make up the action space of the player. Each resource offers a reward to each player who selected the particular resource, where the reward offered depends on the number of players who selected it. The relationship between the reward offered to a player by a resource and the number of users selecting it is captured by the reward function of the resource. A player's utility is equal to the sum of the rewards offered by the resources in the subset selected by the player. In [5], it is established that the above game has a pure-strategy (deterministic) Nash equilibrium.

Although in the classical setting, these games ignore the stochastic nature of the rewards offered by the resources, the idea of resource-sharing games has been extended to different stochastic versions [6,7]. Versions of the game with information asymmetry have been considered through the work of [8] in the context of Bayesian games, which considers the information design problem for resource-sharing with uncertainty. Similar Bayesian games have also been considered in [9,10]. It should be noted that in general resource-sharing games, no conditions are placed on the reward functions of the resources. The special case where the reward functions are non-decreasing in the number of players selecting the resource is called a cost-sharing game [11]. These games are typically treated as games where a cost is minimized rather than a utility being maximized. In fair cost-sharing games, the cost of a resource is divided equally among the players selecting the resource. We consider a fair reward allocation model, where the reward of a resource is equally shared among the players selecting the resource. It should be noted that in this model, the players have opposite incentives compared to a fair-cost sharing model.

The work on resource-sharing games assumes that the players either cooperate or have the incentive to maximize a private or a social utility. It is interesting to consider a stochastic version of the game with asymmetric information between players who do not necessarily trust each other and who place no assumptions on the incentives of the opponents. In this context, the players have no signaling or external feedback and take actions based only on their personal knowledge of the reward realizations for a subset of the resource options. In this paper, we consider the above problem and limit our attention to the two-player singleton case, where each player can choose only one resource.

In the first part of the paper, we provide an iterative best response algorithm to find an $\epsilon$-approximate Nash equilibrium of the system. In the second part, we solve the problem of maximizing the worst-case expected utility of the first player. We solve the problem in two cases. The first case is when both players do not know the realizations of the reward random variables of any of the resources, in which case an explicit solution can be constructed. This case yields a counter-intuitive solution that provides insight into the problem. One such insight is that, while it is always optimal to choose from a subset of resources with the highest average rewards, within that subset, one chooses the higher-valued rewards with lower probability. For the second case, we solve the general version of the problem

by developing an algorithm that leverages the online optimization technique [12,13] and the drift-plus penalty method [14]. This algorithm generates a mixture of $\mathcal{O}(1/\varepsilon^2)$ pure strategies, which, when used in an equiprobable mixture, provides a utility within $\varepsilon$ of optimality on average. Below, we summarize our major contributions.

- We consider the problem of a two-player singleton stochastic resource-sharing game with asymmetric information. We first provide an iterative best response algorithm to find an $\epsilon$-approximate Nash equilibrium of the system. This equilibrium analysis uses potential game concepts.
- When the players do not trust each other and place no assumptions on the incentives of the opponent, we solve the problem of maximizing the worst-case expected utility of the first player using a novel algorithm that leverages techniques from online optimization and the drift-plus penalty methods. The algorithm developed can be used to solve the general unconstrained problem of finding the randomized decision $\alpha \in \{1, 2, \ldots, n\}$, which maximizes $\mathbb{E}\{h(x; \Theta)\}$, where $x \in \mathbb{R}^n$ with $x_k = \mathbb{E}\{\Gamma_k \mathbb{1}_{\{\alpha=k\}}\}$, $\Theta \in \mathbb{R}^m$ and $\Gamma \in \mathbb{R}^n$ are non-negative random vectors with finite second moments, and $h$ is a concave function such that $\tilde{h}(x) = \mathbb{E}\{h(x; \Theta)\}$ is Lipschitz continuous, entry-wise non-decreasing and has bounded subgradients.
- We show our algorithm uses a mixture of only $\mathcal{O}(1/\varepsilon^2)$ pure strategies using a detailed analysis of the sample path of the related virtual queues (our preliminary work on this algorithm used a mixture of $\mathcal{O}(1/\varepsilon^3)$ pure strategies). Virtual queues are also used for constrained online convex optimization in [13], but our problem structure is different and requires a different and more involved treatment.

### 1.1. Background on Resource-Sharing Games

The classical resource-sharing game defined in [5] is a tuple $(\mathcal{M}, \mathcal{N}, \mathcal{T}, r)$, where $\mathcal{M}$ is a set of $m$ players, $\mathcal{N}$ is a set of $n$ resources, $\mathcal{T} = \mathcal{T}_1 \times \mathcal{T}_2 \times \ldots \times \mathcal{T}_m$ where $\mathcal{T}_j$ is the set of possible actions of player $j$ (which is a subset of $2^\mathcal{N}$), and $r = (r_1, r_2, \ldots, r_n)$, where $r_i : \mathbb{N}_0 \to \mathbb{R}$ is the reward function of resource $i$. Here, we use the notation $\mathbb{N}_0 = \mathbb{N} \cup \{0\}$. Each player has complete knowledge about the tuple $(\mathcal{M}, \mathcal{N}, \mathcal{T}, r)$, but they do not have knowledge of the actions chosen by other players. For an action profile $a = (a_1, a_2, \ldots, a_m) \in \mathcal{T}$, the count function # is a function from $\mathcal{N} \times \mathcal{T}$ to $\mathbb{N}_0$ where $\#(i, a) = \sum_{k=1}^m \mathbb{1}_{\{i \in a_k\}}$. In other words, $\#(i, a)$ is the number of players choosing resource $i$ under action profile $a$. We call the quantity $r_i(\#(i, a))$ the *per-player reward* of resource $i$ under action profile $a$. The utility $u_j$ of player $j$ is a function from $\mathcal{T}$ to $\mathbb{R}$, where $u_j(a) = \sum_{i=1}^n \mathbb{1}_{\{i \in a_k\}} r_i(\#(i, a))$. In other words, $u_j(a)$ is the sum of the per-player rewards of the resources chosen by player $j$ under action profile $a$. Resource-sharing games fall under the general category of potential games [15]. Potential games are the class of games for which the change in reward of any player as a result of changing their strategy can be captured by the change in a global potential function.

Many game variations of the resource-sharing game have been studied [16]. Weighted resource-sharing games [17], games with player-dependent reward functions [18], and games with resources having preferences over players [19] are some of the extensions. Singleton games, where each player is allowed to choose only one resource, have also been explored explicitly in the literature [20,21]. Some of the extensions of the classical resource-sharing game possess a pure Nash equilibrium in the singleton case. Two examples would be the games with player-specific reward functions for a resource [18] and the games with priorities where the resources have preferences over the players [19].

Resource-sharing games have been extended to several stochastic versions. For instance, ref. [6] considers the selfish routing problem with risk-averse players in a network with stochastic delays. The work of [7] considers two scenarios where, in the first scenario, each player participates in the game with a certain probability, and in the second scenario, the reward functions are stochastic. The problem of information asymmetry in resource-sharing games has been addressed through the work of [8–10,22]. The work of [22] considers a network congestion game where the players have different information sets

regarding the edges of the network. Further, ref. [8] considers a scenario with a single random state $\theta$, which determines the reward functions. The realization of $\theta$ is known to a game manager who strategically provides recommendations (signaling) to the players to minimize the social cost. An information asymmetry arises among the players in this case due to the actions of the game manager during private signaling, where the game manager provides player-specific recommendations.

Resource-sharing games appear in a variety of applications such as service chain composition [23], congestion control [24], network design [25], load balancing networks [26,27], resource sharing in wireless networks [28], spectrum sharing [29], radio access selection [30], non-orthogonal multiple access [31,32], network selection [33,34], and migration of species [35].

Our formulation differs from the literature on resource-sharing games since we consider a scenario that is difficult to be analyzed using the standard equilibrium-based approaches. This is due to the fact that the players do not trust each other and place no assumptions on the incentives of the opponents, and they take action in the absence of a signaling mechanism or external feedback by just using their knowledge of the reward random variables. This motivates our formulation as a one-shot problem tackled using worst-case expected utility maximization.

*1.2. Notation*

We use calligraphic letters to denote sets. Vectors and matrices are denoted by boldface characters. For integers $n$ and $m$, we denote by $[n:m]$ the inclusive set of integers between $n$ and $m$. Given a vector $\boldsymbol{w} \in \mathbb{R}^m$, $w_k$ is used to denote the $k$-th element of $\boldsymbol{w}$; $\boldsymbol{w}_{k:l}$ for $l \geq k$ represents the $l - k + 1$ dimensional sub-vector $(w_k, w_{k+1}, \ldots, w_l)^\top$ of $\boldsymbol{w}$; for a subset $\mathcal{S}$ of integers from 1 to $n$ $\{w_k; k \in \mathcal{S}\}$ represents the sub-vector of $\boldsymbol{w}$ with index in $\mathcal{S}$. For $\boldsymbol{z} \in \mathbb{R}^m$, we use $\|\boldsymbol{z}\|_2$ to denote the standard Euclidean norm (L2 norm) of $\boldsymbol{z}$. For a function $f : \mathbb{R}^m \to \mathbb{R}$, and $\boldsymbol{z} \in \mathbb{R}^m$, we use $f'(\boldsymbol{z}) = (f'_1(\boldsymbol{z}), f'_2(\boldsymbol{z}), \ldots, f'_m(\boldsymbol{z}))$ to denote a subgradient of $f$ at $\boldsymbol{z}$.

## 2. Materials and Methods

The code used for the simulations is implemented using Python programming language in the notebook https://rb.gy/wvt33, accessed on 10 August 2023.

## 3. Formulation

Denote $\boldsymbol{X} = \{W_k; k \in \mathcal{A}\}$, $\boldsymbol{Y} = \{W_k; k \in \mathcal{B}\}$, $\boldsymbol{Z} = \{W_k; k \in \mathcal{AB}\}$, and $\boldsymbol{V} = \{W_k; k \in \mathcal{C}\}$. Recall that $\boldsymbol{X}$ is known only to player A, $\boldsymbol{Y}$ is known only to player B, $\boldsymbol{Z}$ is known to both players, and $\boldsymbol{V}$ is known to neither. Let us define $\mathcal{A}^c = [1 : n] \setminus \mathcal{A}$, and $\mathcal{B}^c = [1 : n] \setminus \mathcal{B}$. Let $|\mathcal{A}| = a$, $|\mathcal{B}| = b$, $|\mathcal{C}| = c$ and $|\mathcal{AB}| = d$. Therefore, $a + b + c + d = n$. Without loss of generality, we assume $\mathcal{A} = [1 : a]$, $\mathcal{B} = [a + 1 : a + b]$, $\mathcal{C} = [a + b + 1 : a + b + c]$, and $\mathcal{AB} = [a + b + c + 1 : n]$.

Let $R^C(g^A, g^B)$ be the random variable representing the utility of player $C \in \{A, B\}$, given that player A uses strategy $g^A$, and player B uses strategy $g^B$. General strategies for players A and B can be represented by the Borel-measurable functions,

$$g^A : [0, 1) \times \mathbb{R}^{a+d}_{\geq 0} \to [1 : n], \tag{1}$$

$$g^B : [0, 1) \times \mathbb{R}^{b+d}_{\geq 0} \to [1 : n], \tag{2}$$

where

$$\alpha^A = g^A(U^A, \boldsymbol{X}, \boldsymbol{Z}), \tag{3}$$

$$\alpha^B = g^B(U^B, \boldsymbol{Y}, \boldsymbol{Z}), \tag{4}$$

are the resources chosen by players A and B, respectively. Here, $U^A$ and $U^B$ are independent randomization variables uniformly distributed in $[0, 1)$ and independent of $\boldsymbol{W}$. A pure

strategy for player A is a function $g^A$ that does not depend on $U^A$, whereas a mixed strategy is a function $g^A$ that depends on $U^A$. Hence, we drop the randomization variable when depicting a pure strategy. Pure strategies and mixed strategies for player B are defined similarly. Let $\mathcal{S}^A$ and $\mathcal{S}^B$ denote the sets of all possible strategies for players A and B, respectively.

It turns out that our analysis is simplified when $\mathbf{Z}$ is fixed. Fixing $\mathbf{Z}$ does not affect the symmetry between players A and B since $\mathbf{Z}$ is observed by both players A and B. Hereafter, we conduct the analysis by considering all quantities conditioned on $\mathbf{Z}$.

Define

$$p_k^A = \mathbb{E}\{\mathbb{1}_{\{\alpha^A=k\}}|\mathbf{Z}\} \text{ for } 1 \le k \le n,$$
$$q_k^A = \mathbb{E}\{W_k\mathbb{1}_{\{\alpha^A=k\}}|\mathbf{Z}\} \text{ for } k \in \mathcal{A}, \tag{5}$$

and,

$$p_k^B = \mathbb{E}\{\mathbb{1}_{\{\alpha^B=k\}}|\mathbf{Z}\} \text{ for } 1 \le k \le n,$$
$$q_k^B = \mathbb{E}\{W_k\mathbb{1}_{\{\alpha^B=k\}}|\mathbf{Z}\} \text{ for } k \in \mathcal{B}. \tag{6}$$

Note that $p_k^A$ and $p_k^B$ are the conditional probabilities of players A and B choosing $k$ given $\mathbf{Z}$. Define vectors $\boldsymbol{p}^A = \{p_k^A; 1 \le k \le n\}$, $\boldsymbol{q}^A = \{q_k^A; k \in \mathcal{A}\}$, $\boldsymbol{p}^B = \{p_k^B; 1 \le k \le n\}$, and $\boldsymbol{q}^B = \{q_k^B; k \in \mathcal{B}\}$. For $1 \le k \le n$, define $E_k = \mathbb{E}\{W_k|\mathbf{Z}\}$. Hence, we have

$$E_k = \begin{cases} W_k & \text{if } k \in \mathcal{AB}, \\ \mathbb{E}\{W_k\} & \text{otherwise,} \end{cases} \tag{7}$$

which uses the independence of $W_k$ and $\mathbf{Z}$ when $k \notin \mathcal{AB}$.

Note that the utility achieved by player A given the strategies $g^A$ and $g^B$ can be written as

$$R^A(g^A, g^B) = \sum_{k=1}^{n} W_k \left( \mathbb{1}_{\{\alpha^A=k\}} - \frac{1}{2}\mathbb{1}_{\{\alpha^A=k\}}\mathbb{1}_{\{\alpha^B=k\}} \right). \tag{8}$$

Given the strategies $g^A$ and $g^B$, we provide an expression for the expected utility of player A given $\mathbf{Z}$, where the expectation is over the random variables $X, Y, V$, and the possibly random actions $\alpha^A$ and $\alpha^B$. Taking expectations of (8) gives,

$$\mathbb{E}\{R^A(g^A, g^B)|\mathbf{Z}\} = \sum_{k=1}^{n} \mathbb{E}\{W_k\mathbb{1}_{\{\alpha^A=k\}}|\mathbf{Z}\} - \frac{1}{2}\sum_{k=1}^{n} \mathbb{E}\{W_k\mathbb{1}_{\{\alpha^A=k\}}\mathbb{1}_{\{\alpha^B=k\}}|\mathbf{Z}\}$$

$$= \sum_{k\in\mathcal{A}} \mathbb{E}\{W_k\mathbb{1}_{\{\alpha^A=k\}}|\mathbf{Z}\} + \sum_{k\in\mathcal{A}^c} \mathbb{E}\{W_k|\mathbf{Z}\}\mathbb{E}\{\mathbb{1}_{\{\alpha^A=k\}}|\mathbf{Z}\} - \frac{1}{2}\sum_{k=1}^{n} \mathbb{E}\{W_k\mathbb{1}_{\{\alpha^A=k\}}\mathbb{1}_{\{\alpha^B=k\}}|\mathbf{Z}\}$$

$$= \sum_{k\in\mathcal{A}} q_k^A + \sum_{k\in\mathcal{A}^c} E_k p_k^A - \frac{1}{2}\sum_{k=1}^{n} \mathbb{E}\{W_k\mathbb{1}_{\{\alpha^A=k\}}\mathbb{1}_{\{\alpha^B=k\}}|\mathbf{Z}\}. \tag{9}$$

Note that given $\mathbf{Z}$, the random variables $\alpha^A$ and $\alpha^B$ are independent. Hence, we can split the last term (9) as follows,

$$\sum_{k=1}^{n} \mathbb{E}\{W_k\mathbb{1}_{\{\alpha^A=k\}}\mathbb{1}_{\{\alpha^B=k\}}|\mathbf{Z}\} = \sum_{k\in\mathcal{A}} \mathbb{E}\{W_k\mathbb{1}_{\{\alpha^A=k\}}\mathbb{1}_{\{\alpha^B=k\}}|\mathbf{Z}\} + \sum_{k\in\mathcal{B}} \mathbb{E}\{W_k\mathbb{1}_{\{\alpha^A=k\}}\mathbb{1}_{\{\alpha^B=k\}}|\mathbf{Z}\}$$

$$+ \sum_{k\in\mathcal{C}\cup\mathcal{AB}} \mathbb{E}\{W_k\mathbb{1}_{\{\alpha^A=k\}}\mathbb{1}_{\{\alpha^B=k\}}|\mathbf{Z}\} = \sum_{k\in\mathcal{A}} \mathbb{E}\{W_k\mathbb{1}_{\{\alpha^A=k\}}|\mathbf{Z}\}\mathbb{E}\{\mathbb{1}_{\{\alpha^B=k\}}|\mathbf{Z}\}$$

$$+ \sum_{k\in\mathcal{B}} \mathbb{E}\{\mathbb{1}_{\{\alpha^A=k\}}|\mathbf{Z}\}\mathbb{E}\{W_k\mathbb{1}_{\{\alpha^B=k\}}|\mathbf{Z}\} + \sum_{k\in\mathcal{C}\cup\mathcal{AB}} E_k\mathbb{E}\{\mathbb{1}_{\{\alpha^A=k\}}|\mathbf{Z}\}\mathbb{E}\{\mathbb{1}_{\{\alpha^B=k\}}|\mathbf{Z}\}$$

$$= \sum_{k \in \mathcal{A}} q_k^A p_k^B + \sum_{k \in \mathcal{B}} p_k^A q_k^B + \sum_{k \in \mathcal{C} \cup \mathcal{A}\mathcal{B}} E_k p_k^A p_k^B. \tag{10}$$

### 4. Computing the $\epsilon$-Approximate Nash Equilibrium

This section focuses on finding an $\epsilon$-approximate Nash equilibrium of the game. Fix $\epsilon > 0$. A strategy pair $(g^A, g^B)$ is defined as an $\epsilon$-approximate Nash equilibrium if neither player can improve its expected reward by more than $\epsilon$ if it changes its strategy (while holding the strategy of the other player fixed).

Combining (10) with (9), we have that

$$\mathbb{E}\{R^A(g^A, g^B)|\mathbf{Z}\} = \sum_{k \in \mathcal{A}} q_k^A + \sum_{k \in \mathcal{A}^c} E_k p_k^A - \frac{1}{2}\left(\sum_{k \in \mathcal{A}} q_k^A p_k^B + \sum_{k \in \mathcal{B}} p_k^A q_k^B + \sum_{k \in \mathcal{C} \cup \mathcal{A}\mathcal{B}} E_k p_k^A p_k^B\right). \tag{11}$$

Similarly, for player B, we have

$$\mathbb{E}\{R^B(g^A, g^B)|\mathbf{Z}\} = \sum_{k \in \mathcal{B}} q_k^B + \sum_{k \in \mathcal{B}^c} E_k p_k^B - \frac{1}{2}\left(\sum_{k \in \mathcal{A}} q_k^A p_k^B + \sum_{k \in \mathcal{B}} p_k^A q_k^B + \sum_{k \in \mathcal{C} \cup \mathcal{A}\mathcal{B}} E_k p_k^A p_k^B\right). \tag{12}$$

First, we focus on finding the best response for players A and B, given the other player's strategy is fixed.

**Lemma 1.** *The best response for players A and B are given by* $\alpha^A = \arg\max_{1 \le k \le n} A_k$, *and* $\alpha^B = \arg\max_{1 \le k \le n} B_k$, *where $A_k$ and $B_k$ are given by,*

$$A_k = \begin{cases} W_k\left(1 - \frac{p_k^B}{2}\right) & \text{if } k \in \mathcal{A}, \\ E_k - \frac{q_k^B}{2} & \text{if } k \in \mathcal{B}, \\ E_k\left(1 - \frac{p_k^B}{2}\right) & \text{if } k \in \mathcal{C} \cup \mathcal{A}\mathcal{B}, \end{cases} \qquad B_k = \begin{cases} E_k - \frac{q_k^A}{2} & \text{if } k \in \mathcal{A}, \\ W_k\left(1 - \frac{p_k^A}{2}\right) & \text{if } k \in \mathcal{B}, \\ E_k\left(1 - \frac{p_k^A}{2}\right) & \text{if } k \in \mathcal{C} \cup \mathcal{A}\mathcal{B}. \end{cases} \tag{13}$$

**Proof of Lemma 1.** We find the best response for A, and the best response for B follows similarly. Notice that we can rearrange (11) as,

$$\mathbb{E}\{R^A(g^A, g^B)|\mathbf{Z}\} = \sum_{k \in \mathcal{A}} q_k^A\left(1 - \frac{p_k^B}{2}\right) + \sum_{k \in \mathcal{B}} p_k^A\left(E_k - \frac{q_k^B}{2}\right) + \sum_{k \in \mathcal{C} \cup \mathcal{A}\mathcal{B}} p_k^A E_k\left(1 - \frac{p_k^B}{2}\right)$$

$$= \sum_{k \in \mathcal{A}} \mathbb{E}\{W_k \mathbb{1}_{\{\alpha^A=k\}}|\mathbf{Z}\}\left(1 - \frac{p_k^B}{2}\right) + \sum_{k \in \mathcal{B}} \mathbb{E}\{\mathbb{1}_{\{\alpha^A=k\}}|\mathbf{Z}\}\left(E_k - \frac{q_k^B}{2}\right)$$

$$+ \sum_{k \in \mathcal{C} \cup \mathcal{A}\mathcal{B}} E_k \mathbb{E}\{\mathbb{1}_{\{\alpha^A=k\}}|\mathbf{Z}\}\left(1 - \frac{p_k^B}{2}\right) \tag{14}$$

$$= \mathbb{E}\left\{\sum_{k \in \mathcal{A}} W_k\left(1 - \frac{p_k^B}{2}\right)\mathbb{1}_{\{\alpha^A=k\}} + \sum_{k \in \mathcal{B}}\left(E_k - \frac{q_k^B}{2}\right)\mathbb{1}_{\{\alpha^A=k\}} + \sum_{k \in \mathcal{C} \cup \mathcal{A}\mathcal{B}} E_k\left(1 - \frac{p_k^B}{2}\right)\mathbb{1}_{\{\alpha^A=k\}}\,\middle|\,\mathbf{Z}\right\}.$$

The above expectation is maximized when A chooses according to the given policy. □

Next, we find a potential function for the game. A potential function is a function of the strategies of the players such that the change in the utility of a player when he changes his strategy (while the strategies of other players are held fixed) is equal to the change in the potential function [15].

**Theorem 1.** *The function $H(g^A, g^B)$ given by,*

$$H(g^A, g^B) = \sum_{k \in \mathcal{A}}(q_k^A + E_k p_k^B) + \sum_{k \in \mathcal{B}}(q_k^B + E_k p_k^A) + \sum_{k \in \mathcal{C} \cup \mathcal{A}\mathcal{B}} E_k(p_k^A + p_k^B)$$

$$-\frac{1}{2}\left(\sum_{k\in\mathcal{A}}q_k^A p_k^B + \sum_{k\in\mathcal{B}}p_k^A q_k^B + \sum_{k\in\mathcal{C}\cup\mathcal{AB}}E_k p_k^A p_k^B\right), \tag{15}$$

*is a potential function for the game, where $p_k^A$, $p_k^B$ for $1 \le k \le n$, $q_k^A$ for $k \in \mathcal{A}$ and $q_k^B$ for $k \in \mathcal{B}$ are defined in* (5) *and* (6). *Moreover, we have that for all $g^A, g^B \in \mathcal{S}^A \times \mathcal{S}^B$, $H(g^A, g^B) \le 2\sum_{k=1}^n E_k$.*

**Proof of Theorem 1.** The key to the proof is separating (15) (using (11) and (12)) as,

$$H(g^A, g^B) = \mathbb{E}\{R^A(g^A, g^B)\} + \sum_{k\in\mathcal{B}^c}E_k p_k^B + \sum_{k\in\mathcal{B}}q_k^B \tag{16}$$

$$= \mathbb{E}\{R^B(g^A, g^B)\} + \sum_{k\in\mathcal{A}}q_k^A + \sum_{k\in\mathcal{A}^c}p_k^A E_k. \tag{17}$$

Consider updating the strategy of player A while holding the strategy of player B fixed. Notice that since $\sum_{k\in\mathcal{B}^c}E_k p_k^B + \sum_{k\in\mathcal{B}}q_k^B$ is not affected in this process, from (16), we have that the change in the expected utility of player A is equal to the change in the $H$ function. Similarly, this holds when player B updates the strategy while holding player A's strategy fixed. Hence, this is indeed a potential function. The proof that $H(g^A, g^B) \le 2\sum_{k=1}^n E_k$ is omitted for brevity (See technical report [36] for details). □

Using Theorem 1 with standard potential game theory (see, for example, [37]), we have that the iterative best response algorithm with the best response found in Lemma 1 converges to an $\epsilon$-approximate Nash equilibrium in at most $(2\sum_{k=1}^n E_k)/\epsilon$ iterations.

### 5. Worst-Case Expected Utility

Finding a Nash equilibrium using the above algorithm may not be desirable when the players do not trust each other and place no assumptions on the incentives of the opponent. To mitigate this issue, we consider maximizing the worst-case expected utility of player A. Similar to the case of finding the Nash equilibrium, the analysis is simplified when $\mathbf{Z}$ is fixed.

Notice that we can simplify (10) to yield,

$$\sum_{k=1}^n \mathbb{E}\{W_k \mathbb{1}_{\{\alpha^A=k\}}\mathbb{1}_{\{\alpha^B=k\}}|\mathbf{Z}\} \tag{18}$$

$$= \sum_{k\in\mathcal{A}}q_k^A\mathbb{E}\{\mathbb{1}_{\{\alpha^B=k\}}|\mathbf{Z}\} + \sum_{k\in\mathcal{B}}p_k^A\mathbb{E}\{W_k\mathbb{1}_{\{\alpha^B=k\}}|\mathbf{Z}\} + \sum_{k\in\mathcal{C}\cup\mathcal{AB}}E_k p_k^A\mathbb{E}\{\mathbb{1}_{\{\alpha^B=k\}}|\mathbf{Z}\}$$

$$= \sum_{k\in\mathcal{A}}\mathbb{E}\{\Omega_k q_k^A\mathbb{1}_{\{\alpha^B=k\}}|\mathbf{Z}\} + \sum_{k\in\mathcal{A}^c}\mathbb{E}\{\Omega_k p_k^A\mathbb{1}_{\{\alpha^B=k\}}|\mathbf{Z}\}, \tag{19}$$

where

$$\Omega_k = \begin{cases} 1 & \text{if } k \in \mathcal{A}, \\ W_k & \text{if } k \in \mathcal{B}, \\ E_k & \text{if } k \in \mathcal{C} \cup \mathcal{AB}. \end{cases} \tag{20}$$

Plugging the above into (9), we find that

$$\mathbb{E}\{R^A(g^A, g^B)|\mathbf{Z}\} = \sum_{k\in\mathcal{A}}q_k^A + \sum_{k\in\mathcal{A}^c}E_k p_k^A - \frac{1}{2}\mathbb{E}\left\{\sum_{k\in\mathcal{A}}\Omega_k q_k^A\mathbb{1}_{\{\alpha^B=k\}} + \sum_{k\in\mathcal{A}^c}\Omega_k p_k^A\mathbb{1}_{\{\alpha^B=k\}}\,\middle|\,\mathbf{Z}\right\}. \tag{21}$$

The difficulty in dealing with $\mathbb{E}\{R^A(g^A, g^B)|\mathbf{Z}\}$ is that it depends on the strategy $g^B$ of player B, which is not known to player A. Hence, given a strategy $g^A$ of player A, we first focus on obtaining the worst-case strategy $\widehat{g^A}$ of player B. Then we focus on finding the strategy $g^A$ of player A, which maximizes $\mathbb{E}\{R^A(g^A, \widehat{g^A})|\mathbf{Z}\}$. This way, we can guarantee a minimum expected utility for player A irrespective of player B's strategy.

**Lemma 2.** *For given $g^A \in \mathcal{S}^A$, the strategy $g^B \in \mathcal{S}^B$ that minimizes $\mathbb{E}\{R^A(g^A, g^B)|\mathbf{Z}\}$ chooses $\alpha^B = \arg\max_{1 \leq k \leq n} \Lambda_k$, where*

$$\Lambda_k = \begin{cases} \Omega_k q_k^A & \text{if } k \in \mathcal{A}, \\ \Omega_k p_k^A & \text{if } k \in \mathcal{A}^c, \end{cases} \tag{22}$$

*and $\Omega_k$ are defined in (20).*

**Proof of Lemma 2.** Notice that the only term of $\mathbb{E}\{R^A(g^A, g^B)|\mathbf{Z}\}$ in (21) that depends on the strategy of player B is the last expectation. This expectation is maximized when player B chooses $k$, for which $\Lambda_k$ is maximized.[1] $\quad\square$

Hence, we have

$$\mathbb{E}\{R^A(g^A, \widehat{g^A})|\mathbf{Z}\} = \sum_{k \in \mathcal{A}} q_k^A + \sum_{k \in \mathcal{A}^c} E_k p_k^A - \frac{1}{2}\mathbb{E}\{\max\{\Lambda_k; 1 \leq k \leq n\}|\mathbf{Z}\}, \tag{23}$$

where $\Lambda_k$ is defined in (22). We formulate a strategy for player A using the following optimization problem

$$
\begin{aligned}
\text{(P1): } &\underset{g \in \mathcal{S}^A}{\text{maximize}} && f(\mathbf{q}, \mathbf{p}_{a+1:n}) \\
&\text{subject to} && \mathbf{q} \in \mathbb{R}^a, \\
& && \mathbf{p} \in \mathbb{R}^n, \\
& && q_k = \mathbb{E}_{\mathbf{W}, U^A}\{W_k \mathbb{1}_{\{g(U^A, \mathbf{X}, \mathbf{Z})=k\}}|\mathbf{Z}\} \,\forall\, 1 \leq k \leq a, \\
& && p_l = \mathbb{E}_{\mathbf{W}, U^A}\{\mathbb{1}_{\{g(U^A, \mathbf{X}, \mathbf{Z})=l\}}|\mathbf{Z}\} \,\forall\, 1 \leq l \leq n,
\end{aligned}
\tag{24}
$$

where $f : \mathbb{R}^n \to \mathbb{R}$ is defined by,

$$f(\mathbf{x}) = \sum_{k \in \mathcal{A}} x_k + \sum_{k \in \mathcal{A}^c} E_k x_k - \frac{1}{2}\mathbb{E}\{\max\{\Omega_j x_j; 1 \leq j \leq n\}|\mathbf{Z}\}. \tag{25}$$

Although not used immediately, we derive certain properties of $f$ in the following theorem, which are useful later.

**Theorem 2.** *The function $f$*

1. *is concave.*
2. *is entry-wise non-decreasing.*
3. *satisfies,*

$$|f(\mathbf{x}) - f(\mathbf{y})| \leq \frac{3}{2}\sum_{j \in \mathcal{A}} |x_j - y_j| + \frac{3}{2}\sum_{j \in \mathcal{A}^c} E_j|x_j - y_j|, \tag{26}$$

*for any $\mathbf{x}, \mathbf{y} \in \mathbb{R}^n$.*

**Proof of Theorem 2.** See Appendix A. $\quad\square$

It turns out that when $a = b = d = 0$, an explicit solution can be obtained to (P1), which we describe in Section 5.1. In Section 5.2, we describe the solution to the general case. In the technical report [36], we provide simpler alternative solutions to the special cases $a = 0$ (with no restriction on $b$) and $a = 1$ (with the additional assumption that $W_1$ has a continuous CDF).

*5.1. Explicit Solution for a = b = d = 0*

When neither player knows any of the reward realizations, we have $a = b = d = 0$, and the problem reduces to the following.

$$(\text{P2): maximize} \quad \sum_{k=1}^{n} p_k E_k - \frac{1}{2}\max\{p_k E_k; 1 \le k \le n\} \tag{27}$$
$$\text{subject to} \quad \boldsymbol{p} \in \mathcal{I},$$

where

$$\mathcal{I} = \{\boldsymbol{p} \in \mathbb{R}^n : \sum_{i=1}^{n} p_i = 1, p_i \ge 0\, \forall i\} \tag{28}$$

is the *n*-dimensional probability simplex. For this section, we assume without loss of generality that $E_k > 0$ for all $k$. If at least one of the $E_k$'s is zero, we could transform (P2) into a lower dimensional problem with non-zero $E_k$'s. The following lemma constructs an explicit solution for $a = b = d = 0$.[2]

**Lemma 3.** *Assume without loss of generality that $E_k \ge E_{k+1}$ for $1 \le k \le n - 1$. Further, let,*

$$r = \arg\max_{1 \le k \le n} \frac{k - \frac{1}{2}}{\sum_{j=1}^{k} \frac{1}{E_j}}, \tag{29}$$

*where the lowest index is chosen in the case of ties. The optimal solution for (P2) is given by $\boldsymbol{p}^*$ where*

$$p_k^* = \begin{cases} \dfrac{1}{E_k\left(\sum_{j=1}^{r} \frac{1}{E_j}\right)} & \text{if } k \le r, \\ 0 & \text{otherwise.} \end{cases} \tag{30}$$

**Proof of Lemma 3.** See Appendix B. □

It should be noted that this solution is not unique. For instance, consider the case when $n = 2$, $E_1 = 2$, and $E_2 = 1$. In this case, the lemma finds the solution $(p_1, p_2) = (1, 0)$, but it should be noted that $(p_1, p_2) = (1/3, 2/3)$ is also a solution. It is also interesting that the solution assigns positive probabilities to the $r$ resources with the highest average reward, although within these $r$ resources, higher probabilities are assigned to the resources with lower rewards.

It should also be noted that the worst-case strategy can be arbitrarily worse than the Nash equilibrium strategy. For instance, consider the simple scenario with two resources such that $E_1 = E_2$, where none of the players observe any of the reward realizations. In this case, a Nash equilibrium would be player A always choosing resource 1 and player B always choosing resource 2. Another Nash equilibrium would be player B always choosing resource 1 and player A always choosing resource 2. In either case, player A's expected utility is $E_1$. However, notice that, from Lemma 3, the maximum worst-case expected utility of player A is $3E_1E_2/(2E_1 + 2E_2) = 3E_1/4$. Hence, $E_1$ can be scaled to obtain arbitrarily large deviation between the worst-case and the Nash equilibrium solutions.

*5.2. Solving the General Case*

In this section, we focus on solving the most general version of (P1) (with no restrictions on the sets $\mathcal{A}, \mathcal{B}, \mathcal{AB}, \mathcal{C}$). In particular, we focus on finding a mixed strategy to optimize the

worst-case expected utility for player A. It turns out that our optimal solution chooses from a mixture of pure strategies parameterized by $Q \in \mathbb{R}^n$, of the following form

$$g_Q^A(X) = \arg \max_{1 \leq j \leq n} \{\{Q_j W_j; j \in \mathcal{A}\} \cup \{Q_j; j \in \mathcal{A}^c\}\}. \tag{31}$$

We name this special class of pure strategies as *threshold strategies*. We develop a novel algorithm to solve this problem. Our algorithm leverages techniques from drift-plus penalty theory [14] and online convex optimization [12,13]. It should be noted that our algorithm runs offline and is used to construct an appropriate strategy for player A that approximately solves (P1) conditioned on the observed realization of $Z$. We show that we can obtain values arbitrarily close to the optimal value of (P1) by using a finite equiprobable mixture of pure strategies of the above form. It should be noted that the algorithm developed in this section can be used to solve the general unconstrained problem of finding the randomized decision $\alpha \in \{1, 2, \ldots, n\}$ which maximizes $\mathbb{E}\{h(x; \Theta)\}$, where $x \in \mathbb{R}^n$ with $x_k = \mathbb{E}\{\Gamma_k \mathbb{1}_{\{\alpha=k\}}\}$, $\Theta \in \mathbb{R}^m$ and $\Gamma \in \mathbb{R}^n$ are non-negative random vectors with finite second moments, and $h$ is a concave function such that $\tilde{h}(x) = \mathbb{E}\{h(x; \Theta)\}$ is Lipschitz continuous, entry-wise non-decreasing, and has bounded subgradients.

We first provide an algorithm that generates a mixture of $T$ pure strategies, after which we establish the closeness to the optimality of the mixture. We generate a mixture of $T$ pure strategies $\{g_{Q(t)}^A\}_{t=1}^T$ by iteratively updating vector $Q$ for $T$ iterations, where $Q(t)$ and $g_{Q(t)}(X)$ denote the state of $Q$ and the pure strategy generated in the $t$-th iteration, respectively. In addition to $Q(t)$, we require another state vector $\gamma(t) \in \mathbb{R}^n$, which we also update in each iteration, and parameter $V$, which decides the convergence properties of the algorithm. We provide the specific details on setting $V$ later in our analysis. We begin with $Q(1) = \gamma(0) = 0$. In the $t$-th iteration ($t \geq 1$), we independently sample $X(t)$ and $\Omega(t)$ from the distributions of $X$ and $\Omega$, respectively, where $\Omega$ is defined in (20) while keeping $Z$ fixed to its observed value. Then, we update $\gamma(t)$ and $Q(t+1)$ as follows. First, we solve,

$$\text{(P3): } \underset{\gamma(t)}{\text{minimize}} \quad -V f_t'(\gamma(t-1))^\top \gamma(t) + \alpha \|\gamma(t) - \gamma(t-1)\|_2^2 + \sum_{j=1}^n Q_j(t)\gamma_j(t) \tag{32a}$$

$$\text{subject to} \quad \gamma(t) \in \mathcal{K}, \tag{32b}$$

to find $\gamma(t)$, where

$$f_t(x) = \sum_{k \in \mathcal{A}} x_k + \sum_{k \in \mathcal{A}^c} x_k E_k - \frac{1}{2}\max\{x_k \Omega_k(t); 1 \leq k \leq n\}, \tag{33}$$

$\alpha > 0$ and $\mathcal{K} = \left(\times_{j \in \mathcal{A}}[0, E_j]\right) \times [0, 1]^{n-a}$. Notice that $f_t'(x)$ is given by,

$$f_{t,j}'(x) = \begin{cases} 1 - \frac{1}{2}\mathbb{1}_{\{\arg\max_{1 \leq k \leq n}\{x_k \Omega_k(t)\}=j\}} & \text{if } j \in \mathcal{A}, \\ E_j - \frac{1}{2}\mathbb{1}_{\{\arg\max_{1 \leq k \leq n}\{x_k \Omega_k(t)\}=j\}}\Omega_j(t) & \text{if } j \in \mathcal{A}^c, \end{cases} \tag{34}$$

where arg max returns the lowest index in the case of ties. Notice that $f_t$ is a concave function, which can be established by repeating the same argument used to establish the concavity of $f$ in Theorem 2. Then, we choose the action for the $t$-th iteration $\alpha^A(t) = g_{Q(t)}^A(X(t))$ (See (31)). Then, to update $Q(t+1)$, we use,

$$Q_j(t+1) = \max\left\{Q_j(t) + \gamma_j(t) - X_j(t)\mathbb{1}_{\{\alpha^A(t)=j\}}, 0\right\}, \forall j \in \mathcal{A},$$

$$Q_j(t+1) = \max\left\{Q_j(t) + \gamma_j(t) - \mathbb{1}_{\{\alpha^A(t)=j\}}, 0\right\}, \forall j \in \mathcal{A}^c. \tag{35}$$

The algorithm is summarized as Algorithm 1 for clarity.

---

**Algorithm 1:** Algorithm for the generation of the optimal mixture of $T$ pure strategies.

---
1  Initialize $Q(1) = \gamma(0) = 0$
2  **for** *each iteration $t \in [1 : T]$* **do**
3   $\quad$ Sample $X(t)$, and $\Omega(t)$
4   $\quad$ Choose $\gamma(t)$ by solving (P3)
5   $\quad$ Choose the action $\alpha^A(t) = g^A_{Q(t)}(X(t))$
6   $\quad$ Obtain $Q(t+1)$ using (35)
7  **end**

---

After creating the mixture $\{g^A_{Q(t)}\}^T_{t=1}$ of pure strategies, we choose one of them randomly with probability $1/T$ to take the decision. In the following two sections, we focus on solving (P3) and evaluating the performance of Algorithm 1.

5.2.1. Solving (P3)

Notice that the objective of (P3) can be written as

$$- V f'_t(\gamma(t-1))^\top \gamma(t) + \alpha \|\gamma(t) - \gamma(t-1)\|_2^2 + \sum_{j=1}^n Q_j(t) \gamma_j(t)$$

$$= \sum_{j=1}^n \left\{ -V f'_{t,j}(\gamma(t-1)) \gamma_j(t) + \alpha(\gamma_j(t) - \gamma_j(t-1))^2 + Q_j(t) \gamma_j(t) \right\}. \tag{36}$$

Hence (P3) seeks to minimize a separable convex function over the box constraint $\gamma(t) \in \mathcal{K}$. The solution vector $\gamma(t)$ is found by separately minimizing each component $\gamma_j(t)$ over $[0, u_j]$, where

$$u_j = \begin{cases} E_j & \text{if } j \in \mathcal{A}, \\ 1 & \text{if } j \in \mathcal{A}^c. \end{cases} \tag{37}$$

The resulting solution is,

$$\gamma_j(t) = \Pi_{[0, u_j]} \left( \gamma_j(t-1) - \frac{-V f'_{t,j}(\gamma(t-1)) + Q_j(t)}{2\alpha} \right), \tag{38}$$

where $\Pi_{[0, u_j]}$ denotes the projection onto $[0, u_j]$. Notice that the above solution is obtained by projecting the global minimizer of the function to be minimized onto $[0, u_j]$.

5.2.2. How Good Is the Mixed Strategy Generated by Algorithm 1

Without loss of generality, we assume that $E_k > 0$ for all $1 \le k \le n$. The following theorem establishes the closeness of the expected utility generated by Algorithm 1 to the optimal value $f^{\text{opt}}$ of (P1).

**Theorem 3.** *Assume $\alpha$ is set such that $\alpha \ge V^2$, and we use the mixed strategy $g^A$ generated by Algorithm 1 to make the decision. Then,*

$$\mathbb{E}\{R^A(g^A, \widehat{g^A}) | Z\} \ge f^{\text{opt}} - \frac{D_1}{V} - \frac{V D_2}{16\alpha} - \frac{\alpha D_3}{VT} - \frac{3}{2T} \sum_{k \in \mathcal{A}} \left\{ \sqrt{\alpha} + E_k \left( 2\sqrt{2\alpha} + 1 \right) \right\}$$

$$- \frac{3}{2T} \sum_{k \in \mathcal{A}^c} \left\{ E_k^2 \sqrt{\alpha} + E_k \left( 2\sqrt{2\alpha} + 1 \right) \right\}, \tag{39}$$

*where*

$$D_1 = n - a + \frac{1}{2} \sum_{j \in \mathcal{A}} (E_j^2 + \mathbb{E}\{W_k^2\}),$$

$$D_2 = 4a + \mathbb{E}\{\|\mathbf{\Omega}\|_2^2 | \mathbf{Z}\} + \sum_{j \in \mathcal{A}^c} 4E_j^2,$$

$$D_3 = n - a + \sum_{j \in \mathcal{A}} E_j^2, \tag{40}$$

$\mathbf{\Omega}$ *is defined in* (20)*, and* $f^{\mathrm{opt}}$ *is the optimal value of (P1). Hence, by fixing* $\varepsilon > 0$*, and using* $V = 1/\varepsilon$*,* $\alpha = 1/\varepsilon^2$*, and* $T \geq 1/\varepsilon^2$*, the average error is* $\mathcal{O}(\varepsilon)$*.*

**Proof of Theorem 3.** The key to the proof is noticing that $\mathbf{Q}(t)$ can be treated as $n$ queues. Before proceeding with the proof, we define some quantities. Define the history up to time $t$ by $\mathcal{H}(t) = \{\mathbf{X}(\tau); 1 \leq \tau < t\} \cup \{\mathbf{\Omega}(\tau); 1 \leq \tau \leq t\}$. Notice that we include $\mathbf{\Omega}(t)$ in $\mathcal{H}(t)$ since this will allow us to treat $\gamma(t)$ and $\mathbf{Q}(t)$ as deterministic functions of $\mathcal{H}(t)$ and $\mathbf{Z}$. Let us define the Lyapunov function $L(t) = \frac{1}{2}\|\mathbf{Q}(t)\|^2 = \frac{1}{2}\sum_{j=1}^n Q_j(t)^2$, and the drift $\Delta(t) = \mathbb{E}\{L(t+1) - L(t) | \mathcal{H}(t), \mathbf{Z}\}$. Now, notice that

$$\mathbb{E}\{R^A(g^A, \widehat{g^A}) | \mathbf{Z}\} = f\left(\frac{1}{T}\sum_{t=1}^T \mathbb{E}\{\mathbf{x}(t) | \mathbf{Z}\}\right), \tag{41}$$

where

$$x_k(t) = \begin{cases} X_k(t) \mathbb{1}_{\{g_{\mathbf{Q}(t)}^A(\mathbf{X}(t)) = k\}} & \text{if } k \in \mathcal{A}, \\ \mathbb{1}_{\{g_{\mathbf{Q}(t)}^A(\mathbf{X}(t)) = k\}} & \text{if } k \in \mathcal{A}^c. \end{cases} \tag{42}$$

□

We begin with the following two lemmas, which will be useful in the proof.

**Lemma 4.** *The drift is bounded above as*

$$\Delta(t) \leq D_1 + \sum_{j=1}^n Q_j(t)\big(\gamma_j(t) - \mathbb{E}\{x_j(t) | \mathcal{H}(t), \mathbf{Z}\}\big), \tag{43}$$

*where* $D_1$ *is defined in* (40)*.*

**Proof of Lemma 4.** See Appendix C. □

The following is a well-known result regarding the minimization of strongly convex functions (see, for example, a more general pushback result in [38]).

**Lemma 5.** *For a convex function* $h : \mathbb{R}^n \to \mathbb{R}$*, a convex subset* $\mathcal{C}$ *of* $\mathbb{R}^n$*,* $\mathbf{y} \in \mathbb{R}^n$ *and* $\alpha > 0$*, let,*

$$\mathbf{x}^* \in \arg\min_{\mathbf{x} \in \mathcal{C}} \left[h(\mathbf{x}) + \alpha\|\mathbf{x} - \mathbf{y}\|_2^2\right]. \tag{44}$$

*Then,*

$$h(\mathbf{x}^*) + \alpha\|\mathbf{x}^* - \mathbf{y}\|_2^2 \leq h(\mathbf{z}) + \alpha\|\mathbf{z} - \mathbf{y}\|_2^2 - \alpha\|\mathbf{z} - \mathbf{x}^*\|_2^2, \tag{45}$$

*for all* $\mathbf{z} \in \mathcal{C}$*.*

Now, we move on to the main proof. Notice that the objective of (P3) can be written as

$$g'_t(\boldsymbol{\gamma}(t-1))^\top \boldsymbol{\gamma}(t) + \alpha \|\boldsymbol{\gamma}(t) - \boldsymbol{\gamma}(t-1)\|_2^2, \tag{46}$$

where

$$g_t(\boldsymbol{x}) = -Vf_t(\boldsymbol{x}) + \sum_{j=1}^n Q_j(t)x_j. \tag{47}$$

Let $g^{A,*}$ be the strategy that is optimal for (P1). Let us define $\boldsymbol{x}^*(t) \in \mathbb{R}^n$, where

$$x_k^*(t) = \begin{cases} X_k(t)\mathbb{1}_{\{g^{A,*}(U^A(t),\boldsymbol{X}(t),\boldsymbol{Z})=k\}} & \text{if } k \in \mathcal{A}, \tag{48a} \\ \mathbb{1}_{\{g^{A,*}(U^A(t),\boldsymbol{X}(t),\boldsymbol{Z})=k\}} & \text{if } k \in \mathcal{A}^c, \tag{48b} \end{cases}$$

where $U^A(t)$ for $1 \le t \le T$ is a collection of independent and identically distributed uniform $[0,1)$ random variables. Notice that $\boldsymbol{y}^* = \mathbb{E}\{\boldsymbol{x}^*(t)|\boldsymbol{Z}\}$ is independent of $t$ and belongs to $\mathcal{K}$. Hence, $\boldsymbol{y}^*$ is feasible for (P3). Notice that

$$\begin{aligned} &- Vf'_t(\boldsymbol{\gamma}(t-1))^\top \boldsymbol{\gamma}(t) + \sum_{j=1}^n Q_j(t)\gamma_j(t) + \alpha \|\boldsymbol{\gamma}(t) - \boldsymbol{\gamma}(t-1)\|_2^2 \\ &= g'_t(\boldsymbol{\gamma}(t-1))^\top \boldsymbol{\gamma}(t) + \alpha \|\boldsymbol{\gamma}(t) - \boldsymbol{\gamma}(t-1)\|_2^2 \\ &\le_{(a)} g'_t(\boldsymbol{\gamma}(t-1))^\top \boldsymbol{y}^* + \alpha \|\boldsymbol{y}^* - \boldsymbol{\gamma}(t-1)\|_2^2 - \alpha \|\boldsymbol{y}^* - \boldsymbol{\gamma}(t)\|_2^2 \\ &= - Vf'_t(\boldsymbol{\gamma}(t-1))^\top \boldsymbol{y}^* + \sum_{j=1}^n Q_j(t)y_j^* + \alpha \|\boldsymbol{y}^* - \boldsymbol{\gamma}(t-1)\|_2^2 - \alpha \|\boldsymbol{y}^* - \boldsymbol{\gamma}(t)\|_2^2, \end{aligned} \tag{49}$$

where (a) follows from Lemma 5 for the convex function $h$ given by $h(\boldsymbol{x}) = g'_t(\boldsymbol{\gamma}(t-1))^\top \boldsymbol{x}$, and $\mathcal{C} = \mathcal{K}$, since $\boldsymbol{\gamma}(t)$ is the solution to (P3) and $\boldsymbol{y}^*$ is feasible for (P3). Further, step 5 in each iteration of Algorithm 1 of finding the action can be represented as the maximization of

$$\sum_{j \in \mathcal{A}} Q_j(t)\mathbb{E}\{X_j(t)\mathbb{1}_{\{\alpha^A=j\}}|\mathcal{H}(t),\boldsymbol{Z}\} + \sum_{j \in \mathcal{A}^c} Q_j(t)\mathbb{E}\{\mathbb{1}_{\{\alpha^A=j\}}|\mathcal{H}(t),\boldsymbol{Z}\} \tag{50}$$

over all possible actions $\alpha^A \in \{1,2,\dots,n\}$ at time-slot $t$. Hence, comparing the scenario where $g^A_{\boldsymbol{Q}(t)}$ is used in the $t$-th iteration with the scenario where $g^{A,*}$ is used with the randomization variable $U^A(t)$ in the $t$-th iteration, we have the inequality,

$$- \sum_{j=1}^n Q_j(t)\mathbb{E}\{x_j(t)|\mathcal{H}(t),\boldsymbol{Z}\} \le - \sum_{j=1}^n Q_j(t)\mathbb{E}\{x_j^*(t)|\mathcal{H}(t),\boldsymbol{Z}\} = - \sum_{j=1}^n Q_j(t)y_j^*, \tag{51}$$

where the last equality follows since $\boldsymbol{x}^*(t)$ is independent of $\mathcal{H}(t)$. Summing (49) and (51),

$$- Vf'_t(\boldsymbol{\gamma}(t-1))^\top \boldsymbol{\gamma}(t) + \alpha \|\boldsymbol{\gamma}(t) - \boldsymbol{\gamma}(t-1)\|_2^2 + \sum_{j=1}^n Q_j(t)\big(\gamma_j(t) - \mathbb{E}\{x_j(t)|\mathcal{H}(t),\boldsymbol{Z}\}\big) \tag{52}$$

$$\le - Vf'_t(\boldsymbol{\gamma}(t-1))^\top \boldsymbol{y}^* + \alpha \|\boldsymbol{y}^* - \boldsymbol{\gamma}(t-1)\|_2^2 - \alpha \|\boldsymbol{y}^* - \boldsymbol{\gamma}(t)\|_2^2.$$

Adding $D_1 + Vf'_t(\boldsymbol{\gamma}(t-1))^\top \boldsymbol{\gamma}(t-1)$ to both sides and using Lemma 4 yields,

$$\begin{aligned} &\Delta(t) - Vf'_t(\boldsymbol{\gamma}(t-1))^\top \{\boldsymbol{\gamma}(t) - \boldsymbol{\gamma}(t-1)\} + \alpha \|\boldsymbol{\gamma}(t) - \boldsymbol{\gamma}(t-1)\|_2^2 \\ &\le D_1 - Vf'_t(\boldsymbol{\gamma}(t-1))^\top (\boldsymbol{y}^* - \boldsymbol{\gamma}(t-1)) + \alpha \|\boldsymbol{y}^* - \boldsymbol{\gamma}(t-1)\|_2^2 - \alpha \|\boldsymbol{y}^* - \boldsymbol{\gamma}(t)\|_2^2 \\ &\le D_1 - V\{f_t(\boldsymbol{y}^*) - f_t(\boldsymbol{\gamma}(t-1))\} + \alpha \|\boldsymbol{y}^* - \boldsymbol{\gamma}(t-1)\|_2^2 - \alpha \|\boldsymbol{y}^* - \boldsymbol{\gamma}(t)\|_2^2, \end{aligned} \tag{53}$$

where the last inequality follows from the sub-gradient inequality for the concave function $f_t$. Now, we introduce the following lemma.

**Lemma 6.** *We have*

$$-V f_t'(\boldsymbol{\gamma}(t-1))^\top \{\boldsymbol{\gamma}(t) - \boldsymbol{\gamma}(t-1)\} + \alpha \|\boldsymbol{\gamma}(t) - \boldsymbol{\gamma}(t-1)\|_2^2 \geq -\frac{V^2}{4\alpha} \left( a + \sum_{j \in \mathcal{A}^c} E_j^2 \right)$$
$$-\frac{V^2}{16\alpha} \|\boldsymbol{\Omega}(t)\|_2^2, \quad (54)$$

**Proof of Lemma 6.** See Appendix D. □

Substituting the bound from Lemma 6 in (53) we have that

$$\Delta(t) - \frac{V^2}{4\alpha} \left( a + \sum_{j \in \mathcal{A}^c} E_j^2 \right) - \frac{V^2}{16\alpha} \|\boldsymbol{\Omega}(t)\|_2^2$$
$$\leq D_1 - V\{f_t(\boldsymbol{y}^*) - f_t(\boldsymbol{\gamma}(t-1))\} + \alpha \|\boldsymbol{y}^* - \boldsymbol{\gamma}(t-1)\|_2^2 - \alpha \|\boldsymbol{y}^* - \boldsymbol{\gamma}(t)\|_2^2, \quad (55)$$

The above holds for each $t \in \{1, 2, \ldots, T\}$. Hence, we first take the expectation conditioned on $\boldsymbol{Z}$ of both sides of the above expression, after which we sum from 1 to $T$, which results in,

$$\mathbb{E}\{L(T+1)|\boldsymbol{Z}\} - \mathbb{E}\{L(1)|\boldsymbol{Z}\} - \frac{TV^2}{4\alpha} \left( a + \sum_{j \in \mathcal{A}^c} E_j^2 \right) - \frac{TV^2}{16\alpha} \mathbb{E}\{\|\boldsymbol{\Omega}\|_2^2|\boldsymbol{Z}\}$$
$$\leq D_1 T - V \sum_{t=1}^T \mathbb{E}\{f_t(\boldsymbol{y}^*)|\boldsymbol{Z}\} + V \sum_{t=1}^T \mathbb{E}\{f_t(\boldsymbol{\gamma}(t-1))|\boldsymbol{Z}\} + \alpha \mathbb{E}\{\|\boldsymbol{y}^* - \boldsymbol{\gamma}(0)\|_2^2|\boldsymbol{Z}\}$$
$$- \alpha \mathbb{E}\{\|\boldsymbol{y}^* - \boldsymbol{\gamma}(T)\|_2^2|\boldsymbol{Z}\}. \quad (56)$$

Notice that

$$\mathbb{E}\{f_t(\boldsymbol{y}^*)|\boldsymbol{Z}\} = f(\boldsymbol{y}^*) = f^{\text{opt}}, \quad (57)$$

where functions $f$ and $f_t$ are defined in (25) and (33), respectively. Further, we have that

$$\mathbb{E}\{f_t(\boldsymbol{\gamma}(t-1))|\boldsymbol{Z}\} = \mathbb{E}\{\mathbb{E}_{\boldsymbol{\Omega}(t)}\{f_t(\boldsymbol{\gamma}(t-1))|\mathcal{H}(t-1), \boldsymbol{Z}\}|\boldsymbol{Z}\} =_{(a)} \mathbb{E}\{f(\boldsymbol{\gamma}(t-1))|\boldsymbol{Z}\}, \quad (58)$$

where (a) follows from the definition of $f_t$ in (33), since $\boldsymbol{\gamma}(t-1)$ is a function of $\mathcal{H}(t-1)$ and $\boldsymbol{\Omega}(t)$ is independent of $\mathcal{H}(t-1)$. Substituting (57) and (58) into (56), we have that

$$\mathbb{E}\{L(T+1)|\boldsymbol{Z}\} - \mathbb{E}\{L(1)|\boldsymbol{Z}\} - \frac{TV^2}{4\alpha} \left( a + \sum_{j \in \mathcal{A}^c} E_j^2 \right) - \frac{TV^2}{16\alpha} \mathbb{E}\{\|\boldsymbol{\Omega}\|_2^2|\boldsymbol{Z}\}$$
$$\leq D_1 T - VT f^{\text{opt}} + V \sum_{t=1}^T \mathbb{E}\{f(\boldsymbol{\gamma}(t-1))|\boldsymbol{Z}\} + \alpha \mathbb{E}\{\|\boldsymbol{y}^* - \boldsymbol{\gamma}(0)\|_2^2|\boldsymbol{Z}\} - \alpha \mathbb{E}\{\|\boldsymbol{y}^* - \boldsymbol{\gamma}(T)\|_2^2|\boldsymbol{Z}\}$$
$$\leq_{(a)} D_1 T - VT f^{\text{opt}} + V \sum_{t=1}^T \mathbb{E}\{f(\boldsymbol{\gamma}(t-1))|\boldsymbol{Z}\} + \alpha \left( n - a + \sum_{k \in \mathcal{A}} E_k^2 \right)$$
$$\leq D_1 T - VT f^{\text{opt}} + VT f \left( \frac{1}{T} \sum_{t=1}^T \mathbb{E}\{\boldsymbol{\gamma}(t-1)|\boldsymbol{Z}\} \right) + \alpha D_3, \quad (59)$$

where (a) follows since $\boldsymbol{y}^*, \boldsymbol{\gamma}(T), \boldsymbol{\gamma}(0) \in \mathcal{K}$ and the last inequality follows from Jensen's inequality on the concave function $f$. (See the definition of $D_3$ and $D_2$ in (40)). Since $\boldsymbol{Q}(1) = 0$ and $\mathbb{E}\{L(T+1)|\boldsymbol{Z}\} \geq 0$, after some rearrangements above translates to,

$$f^{\mathrm{opt}} - \frac{D_1}{V} - \frac{VD_2}{16\alpha} - \frac{\alpha D_3}{VT} \leq f\left(\frac{1}{T}\sum_{t=0}^{T-1} \mathbb{E}\{\boldsymbol{\gamma}(t)|\boldsymbol{Z}\}\right), \tag{60}$$

where $D_2$ is defined in (40). Now, we prove the following lemma.

**Lemma 7.** *We have*

$$f\left(\frac{1}{T}\sum_{t=0}^{T-1} \mathbb{E}\{\boldsymbol{\gamma}(t)|\boldsymbol{Z}\}\right) \leq f\left(\frac{1}{T}\sum_{t=1}^{T} \mathbb{E}\{\boldsymbol{x}(t)|\boldsymbol{Z}\}\right) + \frac{3}{2T}\sum_{k \in \mathcal{A}}\left\{\sqrt{\alpha} + E_k\left(2\sqrt{2\alpha} + 1\right)\right\}$$

$$+ \frac{3}{2T}\sum_{k \in \mathcal{A}^c}\left\{E_k^2\sqrt{\alpha} + E_k\left(2\sqrt{2\alpha} + 1\right)\right\}. \tag{61}$$

**Proof of Lemma 7.** We first introduce the following two lemmas.

**Lemma 8.** *The queues $Q_j(t)$ for $1 \leq j \leq n$ updated according to Algorithm 1 satisfy,*

$$max\left\{\frac{1}{T}\sum_{t=1}^{T-1} \mathbb{E}\{\gamma_j(t) - x_j(t)|\boldsymbol{Z}\}, \boldsymbol{0}\right\} \leq \frac{\mathbb{E}\{Q_j(T)\,|\boldsymbol{Z}\}}{T}. \tag{62}$$

**Proof of Lemma 8.** See Appendix E.　□

The following lemma is vital in constructing the $\mathcal{O}(\sqrt{\alpha})$ bound on the queue sizes, which leads to the $\mathcal{O}(1/\varepsilon^2)$ solution. It should be noted that an easier bound can be obtained on the queue sizes, which leads to a $\mathcal{O}(1/\varepsilon^3)$ solution.

**Lemma 9.** *Given that $\alpha \geq V^2$, $\boldsymbol{Q}(t)$ satisfy the bound*

$$Q_j(t) \leq \begin{cases} (1 + 2\sqrt{2}E_j)\sqrt{\alpha} + E_j & \text{if } j \in \mathcal{A} \\ (E_j + 2\sqrt{2})\sqrt{\alpha} + 1 & \text{if } j \in \mathcal{A}^c, \end{cases} \tag{63}$$

*for each $t \in [1:T]$.*

**Proof of Lemma 9.** See Appendix F.　□

Now, we move on to the main proof. Notice that

$$f\left(\frac{1}{T}\sum_{t=0}^{T-1} \mathbb{E}\{\boldsymbol{\gamma}(t)|\boldsymbol{Z}\}\right) = f\left(\frac{\boldsymbol{\gamma}(0)}{T} + \frac{1}{T}\sum_{t=1}^{T-1} \mathbb{E}\{\boldsymbol{\gamma}(t)|\boldsymbol{Z}\}\right)$$

$$\leq f\left(\frac{1}{T}\sum_{t=1}^{T} \mathbb{E}\{\boldsymbol{x}(t)|\boldsymbol{Z}\} + \frac{1}{T}\sum_{t=1}^{T-1} \mathbb{E}\{\boldsymbol{\gamma}(t)|\boldsymbol{Z}\} - \frac{1}{T}\sum_{t=1}^{T-1} \mathbb{E}\{\boldsymbol{x}(t)|\boldsymbol{Z}\} - \frac{\mathbb{E}\{\boldsymbol{x}(T)|\boldsymbol{Z}\}}{T}\right)$$

$$\leq_{(a)} f\left(\frac{1}{T}\sum_{t=1}^{T} \mathbb{E}\{\boldsymbol{x}(t)|\boldsymbol{Z}\} + \max\left\{\frac{1}{T}\sum_{t=1}^{T-1} \mathbb{E}\{\boldsymbol{\gamma}(t)|\boldsymbol{Z}\} - \frac{1}{T}\sum_{t=1}^{T-1} \mathbb{E}\{\boldsymbol{x}(t)|\boldsymbol{Z}\}, \boldsymbol{0}\right\}\right) \tag{64}$$

$$\leq_{(b)} f\left(\frac{1}{T}\sum_{t=1}^{T} \mathbb{E}\{\boldsymbol{x}(t)|\boldsymbol{Z}\}\right) + \frac{3}{2}\sum_{k \in \mathcal{A}}\left(\max\left\{\frac{1}{T}\sum_{t=1}^{T-1} \mathbb{E}\{\gamma_k(t) - x_k(t)|\boldsymbol{Z}\}, \boldsymbol{0}\right\}\right)$$

$$+ \frac{3}{2}\sum_{k \in \mathcal{A}^c} E_k\left(\max\left\{\frac{1}{T}\sum_{t=1}^{T-1} \mathbb{E}\{\gamma_k(t) - x_k(t)|\boldsymbol{Z}\}, \boldsymbol{0}\right\}\right),$$

where (a) follows from the entry-wise non-decreasing property of $f$ (Theorem 2-2) and (b) follows from Theorem 2-3. Combining (64) and Lemma 8 with the bound on $Q(T)$ given by Lemma 9, we are finished with the proof of the lemma. $\square$

Combining Lemma 7 with (60), we are finished with the proof of the theorem.

## 6. Simulations

For the simulations, we use $W_j$ as exponential random variables. Notice that since we are conditioning on $\mathbf{Z}$ to solve the problem, the objective of (P1) defined in (25) has the same structure for the two scenarios $(a, b, c, d)$ and $(a, b, c + d, 0)$. Hence, we use $d = 0$ for all the simulations. Notice that the sets $\mathcal{A}$ and $\mathcal{B}$ denote the private information of players A and B, respectively. We consider the three scenarios given below.

1. $a = 0, b = 0, c = 3, d = 0$: Both players do not have private information.
2. $a = 0, b = 1, c = 2, d = 0$: Only player B has private information.
3. $a = 1, b = 1, c = 1, d = 0$: Both players have private information.

Figures 1–3 show pictorial representations of these cases.

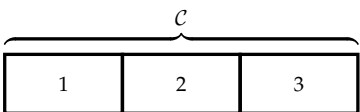

**Figure 1.** $a, b, c, d = 0, 0, 3, 0$.

**Figure 2.** $a, b, c, d = 0, 1, 2, 0$.

**Figure 3.** $a, b, c, d = 1, 1, 1, 0$.

We first consider scenario 1. For Figure 4 (top-left), we fix $E_2 = E_3 = 1$ and plot the expected utilities of players A and B at the $\epsilon$-approximate Nash equilibrium as functions of $E_1$, where $\epsilon = 10^{-3}$ is used. For Figure 4 (top-middle and top-right), we use the same configuration and plot a solution for the probabilities of choosing different resources as a function of $E_1$ at the $\epsilon$-approximate Nash equilibrium for players A and B, respectively. For scenarios 2 and 3, Figure 4 (middle) and Figure 4 (bottom), have similar descriptions to scenario 1.

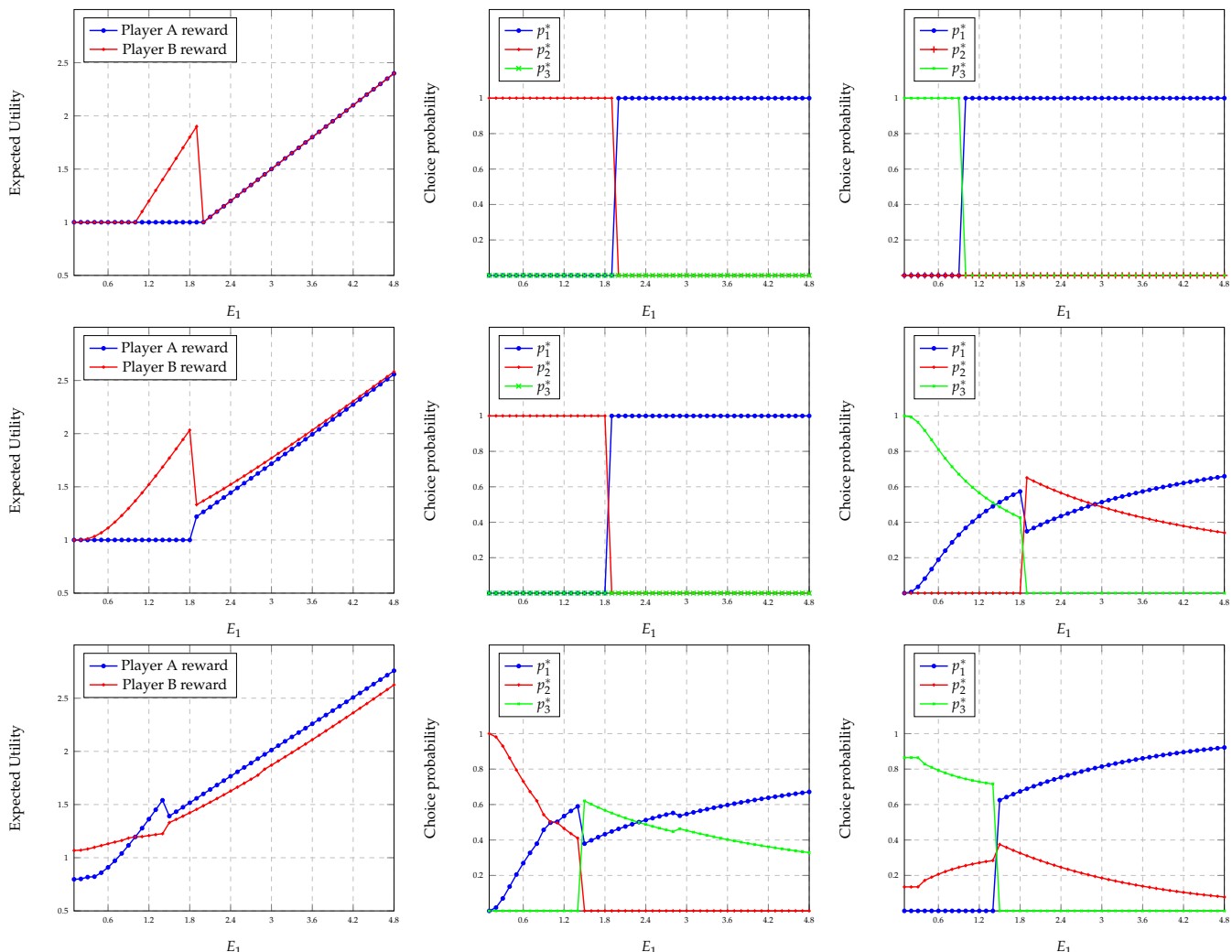

**Figure 4. Top:** Case $a = 0$, $b = 0$, $c = 3$, $d = 0$. **Middle:** Case $a = 0$, $b = 1$, $c = 2$, $d = 0$. **Bottom:** Case $a = b = c = 1$, $d = 0$. **Left:** The expected utility of the players at the $\epsilon$-approximate Nash equilibrium vs. $E_1$. **Middle:** One possible solution for the probabilities of choosing different resources at the $\epsilon$-approximate Nash equilibrium for player A vs. $E_1$. **Right:** One possible solution for the probabilities of choosing different resources at the $\epsilon$-approximate Nash equilibrium for player B vs. $E_1$.

We consider the same three scenarios for the simulations on maximizing the worst-case expected utility. In each scenario, for the top figure, we fix $E_2 = E_3 = 1$ and plot the maximum expected worst-case utility of player A as a function of $E_1$. For the bottom figure, we use the same configuration and plot a solution for the probabilities of choosing different resources for player A as a function of $E_1$. Notice that the solutions may not be unique, as discussed in Section 5.1. Additionally, for Figure 5 (top-middle and top-right), we also indicate the maximum possible error of the solution calculated using the error bound derived in Theorem 3. For scenarios 2 and 3, we have obtained the solutions by averaging over $10^2$ independent simulations. Further, we have used $T = 10^5$, $\alpha = 4 \times 10^4$, and $V = 2 \times 10^2$.

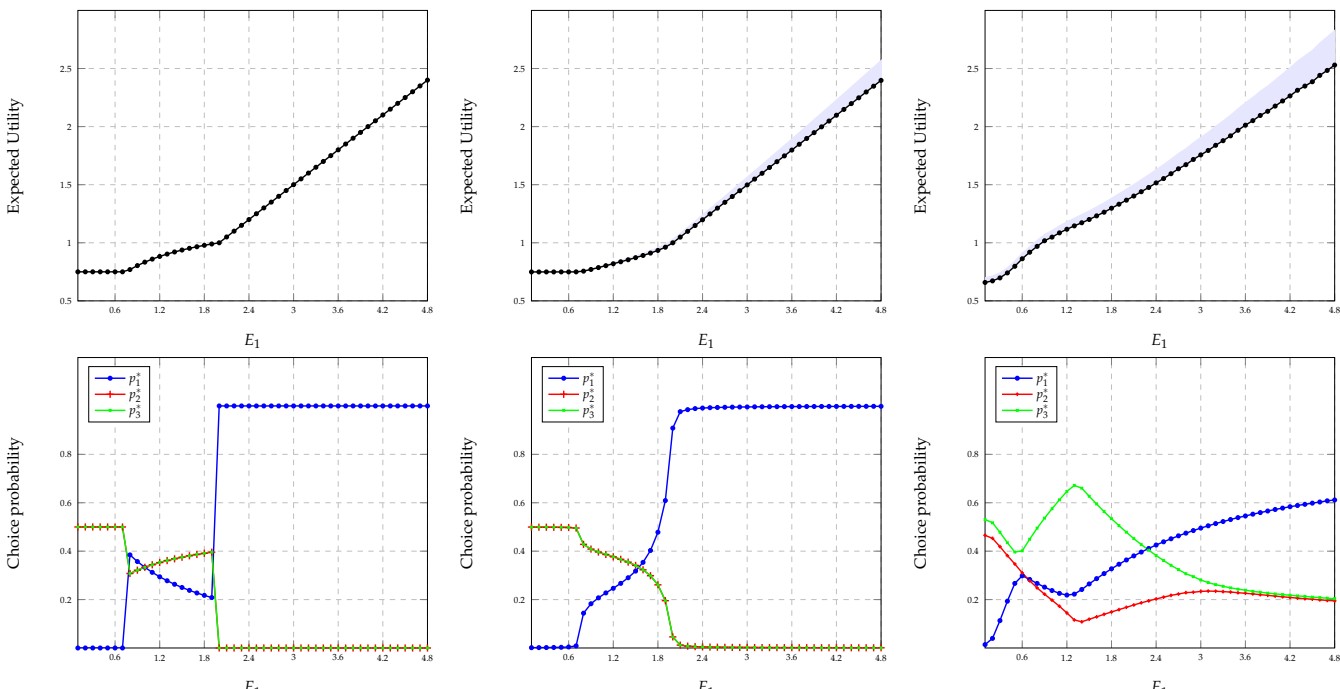

**Figure 5. Left:** Case $a = 0$, $b = 0$, $c = 3$, $d = 0$. **Middle:** Case $a = 0$, $b = 1$, $c = 2$, $d = 0$. **Right:** Case $a = b = c = 1$, $d = 0$. **Top:** The maximum expected worst-case utility of player A and the error margin (shaded in blue) vs. $E_1$. **Bottom:** One possible solution for the probabilities of choosing different resources for player A vs. $E_1$.

Notice that it is difficult to compare the worst-case strategy and the $\epsilon$-approximate Nash equilibrium strategy in general since the first can be computed without any cooperation between the players, whereas computing the second requires cooperation among players. Further, as described in Section 5.1, the worst-case strategy can be arbitrarily worse than the Nash equilibrium strategy. Nevertheless, comparing Figure 4 (left) and Figure 5 (top), it can be seen that the worst-case strategy and the strategy at $\epsilon$-approximate Nash equilibrium yield comparable expected utilities for player A when $E_1 \geq 2$. For instance, in scenario 1, for $E_1 \geq 2$, the approximate Nash equilibrium strategy coincides with the worst-case strategy of choosing resource 1 with probability 1. However, it should be noted that our algorithm for finding the $\epsilon$-approximate Nash equilibrium does not necessarily converge to a socially optimal solution. For instance, in scenario 1, when $E_1 = 2$, player A chooses resource 1 with probability 1 and player B chooses resource 2 with probability 1 gives a higher utility for player A without changing the utility of player B.

In Figure 5, it is interesting to notice the variation in choice probabilities of different resources with $E_1$. Notice that in scenario 1, the choice probability of resource 1 is non-decreasing for $E_1 \in [0.1, 0.8]$, non-increasing for $E_1 \in [0.8, 1.9]$, and non-decreasing for $E_1 \geq 1.9$. Similar behavior can also be observed for scenario 3. This is surprising since intuition suggests that the probability of choosing a resource should increase with the increasing mean of the reward random variable. However, notice that in scenarios 1 and 3, player B does not observe the reward realization of resource 1. This might force player A, playing for the worst case, to believe that player B increases the probability of choosing resource 1 with increasing $E_1$, as a result of which player A chooses resource 1 with a lower probability. Notice that the probability of choosing resource 1 in scenario 3 does not grow as fast as the other two. This is because player A observes $W_1$ and hence can refrain from choosing it when $W_1$ takes low values.

## 7. Conclusions

We have implemented the iterative best response algorithm to find the $\epsilon$-approximate Nash equilibrium of a two-player stochastic resource-sharing game with asymmetric information. To handle situations where the players do not trust each other and place no assumptions on the incentives of the opponent, we solved the problem of maximizing the worst-case expected utility of the first player using a novel algorithm that combines drift-plus penalty theory and online optimization techniques. An explicit solution can be constructed when both players do not observe the realizations of any of the reward random variables. This special case leads to counter-intuitive insights.

In our approach, we have assumed that the reward random variables of different resources are independent. It should be noted that this assumption can be relaxed without affecting the analysis for the special case when both players do not observe the realizations of any of the reward random variables. An interesting question would be what happens in the general case when the reward random variables are not independent. While it is still possible to implement our algorithm in this setting, it is not guaranteed that the algorithm will converge to the optimal solution. Hence, finding an algorithm for this case that exploits the correlations between the reward random variables could be potential future work.

Several other extensions can be considered as well. One would be considering a scenario with multiple players. The general multiplayer case yields a complex information structure since the set of resources has to be split into $2^m$ subsets, where $m$ is the number of players. Additionally, the idea of conditioning on the common information is difficult to be adapted for this case. Nevertheless, various simplified schemes could be considered. One example would be a case with no common information. In this case, the set of resources is split into $m + 1$ disjoint subsets where the $i$-th ($1 \leq i \leq m$) subset is the subset of resources of which the $i$-th player observes the rewards, and the $m + 1$-th subset is the subset of resources of which the rewards are observed by none of the players. Another interesting scenario is when no player observes any of the reward realizations. In both these cases, the expected utility can be calculated following a similar procedure to the two-player case, but finding the worst-case expected utility is difficult. Hence, we believe both cases could be potential future work. Another extension would be extending the algorithm to be implemented with a repeated game structure and in an online scenario.

**Author Contributions:** Conceptualization, M.W. and M.J.N.; methodology, M.W.; software, M.W.; validation, M.W.; writing—original draft preparation, M.W.; writing—review and editing, M.J.N.; visualization, M.W.; supervision, M.J.N.; project administration, M.J.N. All authors have read and agreed to the published version of the manuscript.

**Funding:** This work was supported in part by one or more of: NSF CCF-1718477, NSF SpecEES 1824418.

**Data Availability Statement:** This paper does not use any data from external sources.

**Conflicts of Interest:** The authors declare no conflict of interest.

## Appendix A. Proof of Theorem 2

Notice that the term $\mathbb{E}\{\max\{\Omega_j x_j; 1 \leq j \leq n\}|\mathbf{Z}\}$ of $f$ is convex since the max function is convex and expectation preserves convexity. Hence, $f$ is concave.

For 2 and 3, we use the two inequalities,

$$f(\boldsymbol{x}) - f(\boldsymbol{y}) \geq \sum_{j \in \mathcal{A}} (x_j - y_j) + \sum_{j \in \mathcal{A}^c} E_j(x_j - y_j) - \frac{1}{2}\mathbb{E}\{\max\{\Omega_j(x_j - y_j); j \in [1:n]\}|\mathbf{Z}\}, \tag{A1}$$

and

$$f(\boldsymbol{x}) - f(\boldsymbol{y}) \leq \sum_{j \in \mathcal{A}} (x_j - y_j) + \sum_{j \in \mathcal{A}^c} E_j(x_j - y_j) + \frac{1}{2}\mathbb{E}\{\max\{\Omega_j(y_j - x_j); j \in [1:n]\}|\mathbf{Z}\}, \tag{A2}$$

both of which follow from the fact that for real numbers $\gamma_1, \gamma_2, \gamma_3, \gamma_4$, $\max\{\gamma_1 + \gamma_2, \gamma_3 + \gamma_4\} \leq \max\{\gamma_1, \gamma_3\} + \max\{\gamma_2, \gamma_4\}$.

For 2, we consider $x \geq y$ where the inequality is entry-wise. Notice that

$$
\begin{aligned}
f(x) - f(y) &\geq \sum_{j \in \mathcal{A}} (x_j - y_j) + \sum_{j \in \mathcal{A}^c} E_j(x_j - y_j) - \frac{1}{2}\mathbb{E}\left\{ \sum_{j=1}^{n} \Omega_j (x_j - y_j) \middle| Z \right\} \\
&= \sum_{j \in \mathcal{A}} \frac{1}{2}(x_j - y_j) + \sum_{j \in \mathcal{A}^c} \frac{E_j}{2}(x_j - y_j),
\end{aligned}
$$

where the inequality follows from (A1) and the fact that for $\gamma_1, \gamma_2 \geq 0$, $\max\{\gamma_1, \gamma_2\} \leq \gamma_1 + \gamma_2$.

For 3, note that

$$
\begin{aligned}
f(x) - f(y) &\leq_{(a)} \sum_{j \in \mathcal{A}} |x_j - y_j| + \sum_{j \in \mathcal{A}^c} E_j |x_j - y_j| + \frac{1}{2}\mathbb{E}\left\{ \sum_{j=1}^{n} \Omega_j |x_j - y_j| \middle| Z \right\} \\
&= \frac{3}{2} \sum_{j \in \mathcal{A}} |x_j - y_j| + \frac{3}{2} \sum_{j \in \mathcal{A}^c} E_j |x_j - y_j|,
\end{aligned}
\tag{A3}
$$

where (a) follows from (A2) and the fact that for $\gamma_1, \gamma_2 \geq 0$, $\max\{\gamma_1, \gamma_2\} \leq \gamma_1 + \gamma_2$.

## Appendix B. Proof of Lemma 3

We begin with several results which are used in the proof.

**Lemma A1.** *If $(p^*, \gamma^*)$ solves the problem,*

$$
\begin{aligned}
\text{(P2-1): } \underset{p, \gamma}{\text{maximize}} \quad & \sum_{k=1}^{n} p_k E_k - \frac{1}{2}\gamma \\
\text{subject to} \quad & p \in \mathcal{I}, \\
& \gamma \geq p_k E_k \ \forall 1 \leq k \leq n,
\end{aligned}
\tag{A4}
$$

*where $\mathcal{I}$ is the n-dimensional probability simplex defined in (28), then $p^*$ solves (P2).*

**Proof of Lemma A1.** Define,

$$
f_1(p, \gamma) = \sum_{k=1}^{n} p_k E_k - \frac{1}{2}\gamma.
\tag{A5}
$$

Notice that $f(p) = f_1(p, \max\{p_k E_k; 1 \leq k \leq n\})$. Let $(p^*, \gamma^*)$ be a solution for (P2-1). Notice that for $(p^*, \gamma^*)$ to be feasible for (P2-1), we should have $\gamma^* \geq \max\{p_k^* E_k; 1 \leq k \leq n\}$. However, if $\gamma^* > \max\{p_k^* E_k; 1 \leq k \leq n\}$, we have that $f_1(p, \max\{p_k^* E_k; 1 \leq k \leq n\}) > f_1(p^*, \gamma^*)$, which contradicts the optimality of $(p^*, \gamma^*)$ for (P2-1). Hence, $\gamma^* = \max\{p_k^* E_k; 1 \leq k \leq n\}$. Hence, we have $f(p^*) = f_1(p^*, \gamma^*)$.

Now, consider $\tilde{p} \in \mathcal{I}$. Define $\tilde{\gamma} = \max\{\tilde{p}_k E_k; 1 \leq k \leq n\}$. Since $(\tilde{p}, \tilde{\gamma})$ is also feasible for (P2-1), we should have $f_1(\tilde{p}, \tilde{\gamma}) \leq f_1(p^*, \gamma^*)$. This implies $f(p^*) \geq f(\tilde{p})$. Hence, $p^*$ is an optimal solution of (P2). □

**Lemma A2.** *Consider fixed $\mu \in \mathbb{R}^n$ such that $\mu_k \geq 0$ for all $1 \leq k \leq n$. Now, consider the unconstrained problem,*

$$
\begin{aligned}
\text{(P2-2): } \text{maximize} \quad & f_2(p, \gamma) = \sum_{k=1}^{n} p_k E_k - \frac{1}{2}\gamma + \sum_{k=1}^{n} \mu_k(\gamma - p_k E_k) \\
\text{subject to} \quad & p \in \mathcal{I}, \gamma \in \mathbb{R}.
\end{aligned}
\tag{A6}
$$

*Assume $(p^*, \gamma^*)$ is a solution (P2-2). Additionally, assume that*

$$E_k p_k^* \leq \gamma^* \text{ for all } 1 \leq k \leq n,$$
$$E_k p_k^* = \gamma^* \text{ whenever } \mu_k > 0. \tag{A7}$$

*Then $(p^*, \gamma^*)$ is a solution for (P2-1).*

**Proof of Lemma A2.** First, notice that $(p^*, \gamma^*)$ satisfies the constraints of (P2-1). To show that it maximizes the objective in (P2-1), consider any $(p, \gamma)$ that is feasible for (P2-1). Notice that

$$
\begin{aligned}
f_1(p, \gamma) &= f_2(p, \gamma) - \sum_{\mu_k > 0} \mu_k(\gamma - p_k E_k) \\
&\leq_{(a)} f_2(p^*, \gamma^*) - \sum_{\mu_k > 0} \mu_k(\gamma - p_k E_k) \\
&= f_1(p^*, \gamma^*) + \sum_{\mu_k > 0} \mu_k(\gamma^* - p_k^* E_k - \gamma + p_k E_k) \\
&=_{(b)} f_1(p^*, \gamma^*) - \sum_{\mu_k > 0} \mu_k(\gamma - p_k E_k) \leq_{(c)} f_1(p^*, \gamma^*),
\end{aligned}
\tag{A8}
$$

where $f_1$ is the objective of (P2-1) defined in (A5), $f_2$ is the objective of (P2-2), (a) follows from the optimality of $(p^*, \gamma^*)$ for (P2-2), (b) follows due to (A7), and (c) follows since $\mu_k \geq 0$ and $(p, \gamma)$ is feasible for (P2-1). Hence, we have the result. □

Define,

$$S_k = \sum_{j=1}^{k} \frac{1}{E_j}, \tag{A9}$$

for $1 \leq k \leq n$. We also establish the following lemma, which is useful in our solution.

**Lemma A3.** *Let*

$$r = \arg \max_{1 \leq k \leq n} \frac{k - \frac{1}{2}}{S_k}, \tag{A10}$$

*where $\arg \max$ returns the lowest index in the case of ties. Let us also define $\mu \in \mathbb{R}^n$ as*

$$
\mu_k = \begin{cases} 1 - \frac{1}{E_k} \frac{r - \frac{1}{2}}{S_r} & \text{if } 1 \leq k \leq r, \\ 0 & \text{otherwise.} \end{cases}
\tag{A11}
$$

*Then we have*

1.  $\mu_k \geq 0$ *for all $k$ such that $1 \leq k \leq n$.*
2.  $\sum_{k=1}^{n} \mu_k = \frac{1}{2}$.
3.  $E_k(1 - \mu_k) = \frac{r - \frac{1}{2}}{S_r}$ *for $1 \leq k \leq r$.*
4.  $E_k(1 - \mu_k) \leq \frac{r - \frac{1}{2}}{S_r}$ *for $r + 1 \leq k \leq n$.*

**Proof of Lemma A3.**

1.  Notice that by the definition of $\mu_k$, it is enough to prove the result for $1 \leq k \leq r$. Notice that we are required to prove that

$$\frac{1}{E_k} \frac{r - \frac{1}{2}}{S_r} \leq 1, \tag{A12}$$

for all $1 \le k \le r$. Since $E_k \ge E_{k+1}$ for $1 \le k \le n-1$, it suffices to prove that

$$\frac{1}{E_r} \frac{r - \frac{1}{2}}{S_r} \le 1. \tag{A13}$$

We consider two cases.

**Case 1:** $r = 1$. This case reduces to,

$$\frac{1}{2E_1} \le \frac{1}{E_1}, \tag{A14}$$

which is trivial.

**Case 2:** $r > 1$. Note that from the definition of $r$ in (A10), we have

$$\frac{r - \frac{1}{2}}{S_r} \ge \frac{r - \frac{3}{2}}{S_{r-1}}. \tag{A15}$$

After substituting $S_{r-1} = S_r - \frac{1}{E_r}$ and rearranging, we have the desired result.

2.　Notice that

$$\sum_{k=1}^{n} \mu_k = \sum_{k=1}^{r} \mu_k = \sum_{k=1}^{r} \left(1 - \frac{1}{E_k} \frac{r - \frac{1}{2}}{S_r}\right) = r - \frac{r - \frac{1}{2}}{S_r} \sum_{k=1}^{r} \frac{1}{E_k} = r - \frac{r - \frac{1}{2}}{S_r} S_r = \frac{1}{2}. \tag{A16}$$

3.　This follows from the definition of $\mu_k$ for $1 \le k \le r$.

4.　There is nothing to prove if $r = n$. Hence, we can assume $r < n$. Since $\mu_k = 0$ for $k \ge r+1$, it suffices to prove that $E_k \le \frac{r-(1/2)}{S_r}$. Notice that if we can prove the result for $k = r+1$, we are finished since $E_k \ge E_{k+1}$ for $1 \le k \le n$. Note that from the definition of $r$ in (A10), we have

$$\frac{r - \frac{1}{2}}{S_r} \ge \frac{r + \frac{1}{2}}{S_{r+1}}. \tag{A17}$$

After substituting $S_{r+1} = S_r + \frac{1}{E_{r+1}}$ and rearranging, we have the desired result.

□

Now, we solve the problem using the above lemmas. Consider the problem defined in Lemma A2 with $\mu$ defined in Lemma A3. Specifically, consider the problem,

$$\text{(P2-3): maximize} \quad f_2(\boldsymbol{p}, \gamma) = \sum_{k=1}^{n} p_k E_k - \frac{1}{2}\gamma + \sum_{k=1}^{n} \mu_k(\gamma - p_k E_k)$$

$$\text{subject to} \quad \boldsymbol{p} \in \mathcal{I}, \gamma \in \mathbb{R}. \tag{A18}$$

where $\mu$ and $r$ are defined in (A10) and (A11). For this choice of $\mu_k$ we have

$$f_2(\boldsymbol{p}, \gamma) = \sum_{k=1}^{n} p_k E_k(1 - \mu_k) + \gamma\left(\sum_{k=1}^{n} \mu_k - \frac{1}{2}\right) = \sum_{k=1}^{n} p_k E_k(1 - \mu_k), \tag{A19}$$

where the last equality follows from Lemma A3-2. Now, due to Lemma A3-3 and Lemma A3-4, the optimal solution for (P2-3) is any $(\boldsymbol{p}, \gamma)$ such that $\gamma \in \mathbb{R}$, and $\boldsymbol{p} \in \mathcal{I}$ such that $p_k = 0$ for $k > r$. In particular, consider the solution $(\boldsymbol{p}^*, \gamma^*)$ given by,

$$p_k^* = \begin{cases} \frac{1}{E_k S_r} & \text{if } k \le r, \\ 0 & \text{otherwise,} \end{cases} \tag{A20}$$

and $\gamma^* = \frac{1}{S_r}$. Notice that for $1 \leq k \leq r$, we have that $p_k^* E_k = \gamma^*$, and $p_k^* E_k = 0 \leq \gamma^*$ for $r + 1 \leq k \leq n$. Hence, from Lemma A2, $(p^*, \gamma^*)$ is a solution for (P2-1). Hence, from Lemma A1, $p^*$ is a solution for (P2) as desired.

**Appendix C. Proof of Lemma 4**

Notice that

$$\Delta(t) = \mathbb{E}\{L(t+1) - L(t)|\mathcal{H}(t), Z\} = \frac{1}{2}\mathbb{E}\left\{\sum_{j=1}^n Q_j(t+1)^2 - Q_j(t)^2 \bigg| \mathcal{H}(t), Z\right\}$$

$$= \frac{1}{2}\sum_{j=1}^n \mathbb{E}\{Q_j(t+1)^2|\mathcal{H}(t), Z\} - \frac{1}{2}\sum_{j=1}^n Q_j(t)^2$$

$$= \frac{1}{2}\sum_{j=1}^n \mathbb{E}\{\max\{Q_j(t) + \gamma_j(t) - x_j(t), 0\}^2|\mathcal{H}(t), Z\} - \frac{1}{2}\sum_{j=1}^n Q_j(t)^2$$

$$\leq \frac{1}{2}\sum_{j=1}^n \mathbb{E}\{(Q_j(t) + \gamma_j(t) - x_j(t))^2|\mathcal{H}(t), Z\} - \frac{1}{2}\sum_{j=1}^n Q_j(t)^2$$

$$\leq \sum_{j=1}^n Q_j(t)\big(\gamma_j(t) - \mathbb{E}\{x_j(t)|\mathcal{H}(t), Z\}\big) + \frac{1}{2}\sum_{j=1}^n \gamma_j(t)^2 + \frac{1}{2}\sum_{j=1}^n \mathbb{E}\{x_j(t)^2|\mathcal{H}(t), Z\}$$

$$\leq_{(a)} \sum_{j=1}^n Q_j(t)\big(\gamma_j(t) - \mathbb{E}\{x_j(t)|\mathcal{H}(t), Z\}\big) + \frac{1}{2}\sum_{j\in\mathcal{A}} E_j^2 + \frac{n-a}{2}$$

$$+ \frac{1}{2}\sum_{j\in\mathcal{A}} \mathbb{E}\{(X_j(t)\mathbb{1}_{\{\alpha(t)=k\}})^2|\mathcal{H}(t), Z\} + \frac{1}{2}\sum_{j\in\mathcal{A}^c} \mathbb{E}\{(\mathbb{1}_{\{\alpha(t)=k\}})^2|\mathcal{H}(t), Z\}$$

$$\leq \sum_{j=1}^n Q_j(t)\big(\gamma_j(t) - \mathbb{E}\{x_j(t)|\mathcal{H}(t), Z\}\big) + \frac{1}{2}\sum_{j\in\mathcal{A}} E_j^2 + \frac{n-a}{2} + \frac{1}{2}\sum_{j\in\mathcal{A}} \mathbb{E}\{X_j(t)^2|\mathcal{H}(t), Z\}$$

$$+ \frac{1}{2}\sum_{j\in\mathcal{A}^c} \mathbb{E}\{1|\mathcal{H}(t), Z\}$$

$$=_{(b)} \sum_{j=1}^n Q_j(t)\big(\gamma_j(t) - \mathbb{E}\{x_j(t)|\mathcal{H}(t), Z\}\big) + \frac{1}{2}\sum_{j\in\mathcal{A}} E_j^2 + \frac{n-a}{2} + \frac{1}{2}\sum_{j\in\mathcal{A}} \mathbb{E}\{W_j^2\} + \frac{n-a}{2}$$

$$= \sum_{j=1}^n Q_j(t)\big(\gamma_j(t) - \mathbb{E}\{x_j(t)|\mathcal{H}(t), Z\}\big) + D_1, \tag{A21}$$

where inequality (a) follows since $\gamma(t) \in \mathcal{K}$ and equality (b) follows from the fact that $X(t)$ is independent of $\mathcal{H}(t)$ and $Z$.

**Appendix D. Proof of Lemma 6**

Notice that

$$f_t'(\gamma(t-1)) = v - \frac{1}{2}\tilde{\Omega}(t), \tag{A22}$$

where $v$ is defined by,

$$v_j = \begin{cases} 1 & \text{if } j \in \mathcal{A}, \\ E_j & \text{if } j \in \mathcal{A}^c. \end{cases} \tag{A23}$$

$\tilde{\Omega}(t)$ is given by $\tilde{\Omega}_k(t) = \Omega_k(t)\mathbb{1}_{\{\arg\max_{1\leq j\leq n}\{\gamma_j(t-1)\Omega_j(t)\}=k\}}$, and arg max returns the least index in the case of ties. Notice that

$$- Vf_t'(\gamma(t-1))^\top \{\gamma(t) - \gamma(t-1)\} + \alpha\|\gamma(t) - \gamma(t-1)\|_2^2$$

$$\geq_{(a)} -V\|f_t'(\boldsymbol{\gamma}(t-1))\|_2\|\boldsymbol{\gamma}(t)-\boldsymbol{\gamma}(t-1)\|_2 + \alpha\|\boldsymbol{\gamma}(t)-\boldsymbol{\gamma}(t-1)\|_2^2$$

$$= \alpha\left(\|\boldsymbol{\gamma}(t)-\boldsymbol{\gamma}(t-1)\|_2 - \frac{V}{2\alpha}\|f_t'(\boldsymbol{\gamma}(t-1))\|_2\right)^2 - \frac{V^2}{4\alpha}\|f_t'(\boldsymbol{\gamma}(t-1))\|_2^2$$

$$\geq -\frac{V^2}{4\alpha}\|f_t'(\boldsymbol{\gamma}(t-1))\|_2^2 = -\frac{V^2}{4\alpha}\left\|\boldsymbol{v}-\frac{1}{2}\tilde{\boldsymbol{\Omega}}(t)\right\|_2^2 \geq_{(b)} -\frac{V^2}{4\alpha}\|\boldsymbol{v}\|_2^2 - \frac{V^2}{16\alpha}\|\tilde{\boldsymbol{\Omega}}(t)\|_2^2$$

$$\geq -\frac{V^2}{4\alpha}\left(a+\sum_{j\in\mathcal{A}^c}E_j^2\right) - \frac{V^2}{16\alpha}\|\boldsymbol{\Omega}(t)\|_2^2, \tag{A24}$$

where (a) follows from the Cauchy–Schwarz inequality, (b) follows since $v_k \geq 0$ and $\tilde{\Omega}_k(t) \geq 0$ for all $1 \leq k \leq n$.

### Appendix E. Proof of Lemma 8

Notice that from the definition of $Q_j(t+1)$ in (35) and the definition of $x_j(t)$ in (42) we have that

$$Q_j(t+1) - Q_j(t) \geq \gamma_j(t) - x_j(t), \tag{A25}$$

for all $1 \leq j \leq n$ and $1 \leq t \leq T-1$. Summing the above from 1 to $T-1$, we have that

$$Q_j(T) - Q_j(1) \geq \sum_{t=1}^{T-1}\{\gamma_j(t) - x_j(t)\}. \tag{A26}$$

After using $Q_j(1) = 0$, taking expectations conditioned on $\boldsymbol{Z}$ and some algebraic manipulations, we have

$$\frac{\mathbb{E}\{Q_j(T)|\boldsymbol{Z}\}}{T} \geq \frac{1}{T}\sum_{t=1}^{T-1}\mathbb{E}\{\gamma_j(t) - x_j(t)|\boldsymbol{Z}\}. \tag{A27}$$

We have the desired inequality from the above since $Q_j(T)$ is non-negative.

### Appendix F. Proof of Lemma 9

Define $\boldsymbol{v}, \boldsymbol{u}$ as follows.

$$v_k = \begin{cases} 1 & \text{if } k \in \mathcal{A}, \\ E_k & \text{if } k \in \mathcal{A}^c, \end{cases} \quad \text{and} \quad u_k = \begin{cases} E_k & \text{if } k \in \mathcal{A}, \\ 1 & \text{if } k \in \mathcal{A}^c. \end{cases} \tag{A28}$$

Hence we are required to prove that $Q_j(t) \leq (v_j + 2\sqrt{2}u_j)\sqrt{\alpha} + u_j$ for all $t \in [1:T]$. We begin with several important results.

**Lemma A4.** *We have the following results regarding $Q_j(t)$.*

1.  *$Q_j(t+1) \leq Q_j(t) + u_j$ for all $t \geq 1$.*
2.  *Assume $Q_j(t) \geq (v_j + \sqrt{2}u_j)\sqrt{\alpha}$ for some $t \geq 1$. Then we have either $\gamma_j(t) = 0$ or*

$$\gamma_j(t) \leq \gamma_j(t-1) - \frac{u_j}{\sqrt{2\alpha}}. \tag{A29}$$

3.  *Assume $Q_j(\tau) \geq (v_j + \sqrt{2}u_j)\sqrt{\alpha}$ for all $\tau \in [t:t+t_0]$, where $t \geq 1$ and $t_0 \geq 0$. Additionally assume $\gamma_j(t-1) = 0$. Then $\gamma_j(\tau) = 0$ for all $\tau \in [t-1:t+t_0]$.*

**Proof of Lemma A4.**

1.  Notice that from the definition of $Q_j(t+1)$ in (35), for $j \in \mathcal{A}$ we have

$$Q_j(t+1) = \max\left\{Q_j(t) + \gamma_j(t) - X_j(t)\mathbb{1}_{\{\alpha^A(t)=j\}}, 0\right\} \le \max\{Q_j(t) + u_j, 0\}$$
$$= Q_j(t) + u_j, \tag{A30}$$

where the inequality follows from the definition of $u_j$ in (A28). The same argument can be repeated for $j \in \mathcal{A}^c$.

2. Notice that if $\gamma_j(t) \ne 0$ then we have

$$\gamma_j(t) \le \gamma_j(t-1) - \frac{-Vf'_{t,j}(\gamma(t-1)) + Q_j(t)}{2\alpha}, \tag{A31}$$

which follows since $\gamma_j(t)$ is the projection of $\gamma_j(t-1) - \frac{-Vf'_{t,j}(\gamma(t-1))+Q_j(t)}{2\alpha}$ onto $[0, u_j]$ (See (38)). Hence, we have that

$$\gamma_j(t) \le \gamma_j(t-1) - \frac{-Vf'_{t,j}(\gamma(t-1)) + Q_j(t)}{2\alpha}$$
$$\le_{(a)} \gamma_j(t-1) - \frac{-Vv_j + (v_j + \sqrt{2}u_j)\sqrt{\alpha}}{2\alpha} \le_{(b)} \gamma_j(t-1) - \frac{u_j}{\sqrt{2\alpha}}, \tag{A32}$$

where (a) follows from the subgradients of $f_t$ found in (34) and (b) follows from $\alpha \ge V^2$.

3. Notice if we prove $\gamma_j(t) = 0$, we can use the same argument inductively to establish the result. Assume the contrary that $\gamma_j(t) \ne 0$. Then, from part 2, we should have

$$\gamma_j(t) \le \gamma_j(t-1) - \frac{u_j}{\sqrt{2\alpha}} = -\frac{u_j}{\sqrt{2\alpha}}, \tag{A33}$$

which is a contradiction since $\gamma_j(t) \ge 0$. Hence, we have the result.

□

Now, we use an inductive argument to prove the main result. Notice that the result is true for $t = 1$, since $Q_j(1) = 0 \le (v_j + 2\sqrt{2}u_j)\sqrt{\alpha} + u_j$. Now, we prove that $Q_j(t+1) \le (v_j + 2\sqrt{2}u_j)\sqrt{\alpha} + u_j$ for $t \ge 1$, with the assumption that $Q_j(t) \le (v_j + 2\sqrt{2}u_j)\sqrt{\alpha} + u_j$.

We consider three cases.

**Case 1:** $Q_j(t) \le (v_j + 2\sqrt{2}u_j)\sqrt{\alpha}$. This case follows from Lemma A4-1.

**Case 2:** $t \le \sqrt{2\alpha} + 1$. Notice that

$$Q_j(t+1) \le Q_j(1) + u_j t \le (\sqrt{2\alpha} + 1)u_j \le (v_j + 2\sqrt{2}u_j)\sqrt{\alpha} + u_j, \tag{A34}$$

where the first inequality follows from Lemma A4-1.

**Case 3:** $t > \sqrt{2\alpha} + 1$ and $Q_j(t) > (v_j + 2\sqrt{2}u_j)\sqrt{\alpha}$. For this, we prove that $\gamma_j(t) = 0$, which establishes the claim from the definition of $Q_j(t+1)$ in (35) and the induction hypothesis.

Notice that for all $u \in [1:t]$ we have

$$Q_j(u) \ge_{(a)} Q_j(t) - (t-u)u_j \ge (v_j + 2\sqrt{2}u_j)\sqrt{\alpha} - (t-u)u_j$$
$$= (v_j + \sqrt{2}u_j)\sqrt{\alpha} + \sqrt{2\alpha}u_j - (t-u)u_j, \tag{A35}$$

where (a) follows from Lemma A4-1.

Hence, for all $u \in \mathbb{Z}$ such that $t - \sqrt{2\alpha} \le u \le t$, we have that

$$Q_j(u) \ge (v_j + \sqrt{2}u_j)\sqrt{\alpha}. \tag{A36}$$

Now, we prove that there exists $u \in \mathbb{Z}$ such that $t - \sqrt{2\alpha} \le u \le t$ and $\gamma_j(u) = 0$, which will establish that $\gamma_j(t) = 0$ from Lemma A4-3. For the proof, assume the contrary

$\gamma_j(u) > 0$ for all $u \in \mathbb{Z}$ such that $t - \sqrt{2\alpha} \le u \le t$ (or $u \in [t - \lfloor \sqrt{2\alpha} \rfloor : t]$, where $\lfloor x \rfloor$ denotes the largest integer smaller than or equal to $x$). From Lemma A4-2, we have that

$$\gamma_j(t) \le \gamma_j\left(t - \lfloor \sqrt{2\alpha} \rfloor - 1\right) - \frac{\left(\lfloor \sqrt{2\alpha} \rfloor + 1\right) u_j}{\sqrt{2\alpha}} \le 0, \tag{A37}$$

where the last inequality follows since $\lfloor x \rfloor + 1 \ge x$ and $\gamma_j\left(t - \lfloor \sqrt{2\alpha} \rfloor - 1\right) \le u_j$ (since $\gamma_j(\tau) \in [0, u_j]$ for all $\tau \in [1 : T]$ by the projection definition of $\gamma_j(\tau)$ in (38), and $\gamma_j(0) = 0$). Hence, we should have that $\gamma_j(t) = 0$, which contradicts our initial assumption. Hence, we are finished.

## Notes

[1] Ideally, player B may not have information about $q_j^A$ and $p_j^A$. Hence, player B may not be able to utilize this exact strategy. Nevertheless, obtaining a better bound is impossible since we do not have any assumptions or information about player B's strategy. For instance, if player B assumes that player A is using a particular strategy and if player B's assumption turns out to be correct since player B knows the distributions of all $W_j$ for $1 \le j \le n$, player B's estimates of $q_j^A$ and $p_j^A$ are exact.

[2] The same problem structure arises in the case with symmetric information between the players (case $a = b = 0$ with $d$ arbitrary). Hence, we can use the solution obtained in this section for the above case as well.

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
