# Peer review of "A Two-Player Resource-Sharing Game with Asymmetric Information"

_games, doi:10.3390/g14050061_

Round 1

Reviewer 1 Report

This paper is on a two-person resource sharing game context. There sare two bidders that can bid and obtain random rewards from resources that they they choose. The key part of the paper is on the information structures available for the two players. A subset of resources can be viewed by either one of the players and another subset can be observed by both and a final subset cannot be viewed by either. The goal is to maximize the average rewards. It is assumed that the reward distributions are known and are i.i.d.

The paper consists of two principal results: the first is the demonstration of an epsilon Nash equilibrium by studying the best responses of each player and then showing the existence of a potential function.

The second and major part of the paper is based on the situation where the rewards of the other player are unknown and so the idea is to study the worst case utilities (rewards) for each player. In some trivial special cases when neither player has any information the paper shows that it is possible to compute the equilibrium explicitly and exhibits an interesting property related to that the optimal policy attaches positive probabilities to the r highest rewards but chooses the least likely amongst them. The auhtors show how this problem can be solved by a Lyapunov gradient method and then present some simulation results to compare the worst case approach with the epsilon-optimal solutions.

Overall the paper is well presented and easy to follow. The analysis is rigorous. However, I do have some comments.

1) How does the analysis carry over when there are more than 2 agents? I think the existence of epsilon-optimal Nash equilibria would hold. What would change with respect to the second approach since now the worst-case would be much more difficult to characterize and hence structurally the gradient based approach might not be easy to develop because convexity might be lost. Some comments on this would be welcome.

2) There is a problem in equation (24). The definition of q_k should not involve the index l.

3) The fact that the worst case does not differ from the epsilon Nash might be more due to the assumptions. Can the authors give bounds on how much they will differ based on the statistics of the rewards?

Overall I found the paper interesting and I recommend it for publication.

Reviewer 2 Report

The reviewed article introduces a novel algorithm that merges drift-plus penalty theory with online optimization techniques to tackle the challenging problem of maximizing the worst-case expected utility for a player. This contribution significantly advances the realms of algorithms and game theory, offering intriguing insights that challenge conventional research directions. Nevertheless, while the article is commendable, several areas require improvement for a more comprehensive and practical contribution to the field.

Visual Consistency and Parameter Explanation:

In Figure 5, it is imperative to ensure visual consistency between E1 and E2, E3 to enhance the article's readability and clarity. Additionally, it is advisable to provide a thorough explanation for the chosen parameter values (a, b, d) to justify their reasonability. This will help readers better understand the significance of these parameters within the context of the algorithm.

Real-Life Examples in the Introduction:

To make the research questions more relatable and practical, the introductory section should incorporate relevant real-life examples. This addition will not only enhance the article's accessibility but also prevent it from appearing overly abstract or disconnected from practical applications. Real-world scenarios can serve as compelling motivators for readers to engage with the research.

Expanding Algorithm Extensions:

Consider expanding the algorithm to accommodate various extensions that relax the conditions for its practical application. This would enhance the algorithm's versatility and potential real-world applicability, making it more valuable to researchers and practitioners in the field. Exploring these extensions could also provide additional insights and contributions to the research.

Elaborating Conclusions:

The conclusions section currently offers concise propositions. To improve the completeness and comprehensibility of the article's conclusions, it is recommended to expand this section. Providing more detailed explanations, discussing implications, and suggesting avenues for future research will enhance the overall impact of the article and help readers better grasp the significance of the findings.

In summary, the reviewed article presents a novel algorithm with considerable potential in the realms of algorithms and game theory. To further enhance its value and accessibility, improvements in visual consistency, parameter explanation, real-life examples, algorithm extensions, and conclusions elaboration are suggested. These enhancements will contribute to a more robust and impactful research article that resonates with both scholars and practitioners in the field.
